# ImOV3D: Learning Open-Vocabulary Point Clouds 3D Object Detection from Only 2D Images

**Timing Yang**[1,2*]  **Yuanliang Ju**[1,2*]  **Li Yi**[2,3,1†]

[1] Shanghai Qi Zhi Institute, [2] IIIS, Tsinghua University, [3] Shanghai AI Lab

## Abstract

Open-vocabulary 3D object detection (OV-3Det) aims to generalize beyond the limited number of base categories labeled during the training phase. The biggest bottleneck is the scarcity of annotated 3D data, whereas 2D image datasets are abundant and richly annotated. Consequently, it is intuitive to leverage the wealth of annotations in 2D images to alleviate the inherent data scarcity in OV-3Det. In this paper, we push the task setup to its limits by exploring the potential of using solely 2D images to learn OV-3Det. The major challenges for this setup is the modality gap between training images and testing point clouds, which prevents effective integration of 2D knowledge into OV-3Det. To address this challenge, we propose a novel framework **ImOV3D** to leverage pseudo multimodal representation containing both images and point clouds (PC) to close the modality gap. The key of ImOV3D lies in flexible modality conversion where 2D images can be lifted into 3D using monocular depth estimation and can also be derived from 3D scenes through rendering. This allows unifying both training images and testing point clouds into a common image-PC representation, encompassing a wealth of 2D semantic information and also incorporating the depth and structural characteristics of 3D spatial data. We carefully conduct such conversion to minimize the domain gap between training and test cases. Extensive experiments on two benchmark datasets, SUNRGBD and ScanNet, show that ImOV3D significantly outperforms existing methods, even in the absence of ground truth 3D training data. With the inclusion of a minimal amount of real 3D data for fine-tuning, the performance also significantly surpasses previous state-of-the-art. Codes and pre-trained models are released on the `https://github.com/yangtiming/ImOV3D`.

## 1 Introduction

In the 3D vision community, there is a notable surge in interest surrounding open-vocabulary 3D object detection (OV-3Det). This task focuses on the detection of objects from unbounded categories that were not present during the training phase, using 3D point clouds as input. Such capability holds immense significance in dynamic 3D environments where a wide range of object categories constantly emerge and evolve, which is critical in downstream applications including robotics[8, 18, 28, 23], autonomous driving [31, 46], and augmented reality [35, 44].

With the advancements in OV-3Det, which is not only scarce in terms of labels but also in the data itself. However, the collection and annotation of 3D point clouds scenes pose significant challenges. The availability of accessible and scannable scenes *(e.g. indoor scenes)* may be limited. Additionally, obtaining 3D annotations often requires substantial human effort and time-consuming. These limitations impact the model's performance in handling novel objects. Existing methods [30, 29, 28, 10, 49] seek help from powerful open-vocabulary 2D detectors. A common method

---

[*]Equal contribution.

[†]Corresponding author.

38th Conference on Neural Information Processing Systems (NeurIPS 2024).

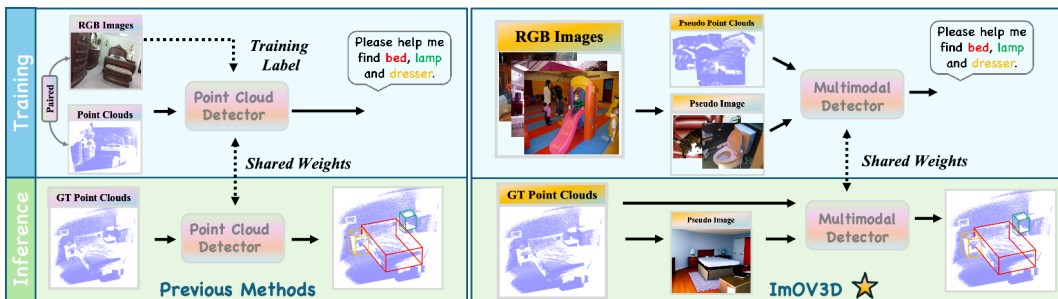

Figure 1: **Left:** Traditional methods require paired RGB-D data for training and use single-modality point clouds as input during inference. **Right:** ImOV3D involves using a vast amount of 2D images to generate pseudo point clouds during the training phase, which are then rendered back into images. In the inference phase, with only point clouds as input, we still construct a pseudo-multimodal representation to enhance detection performance.

leverages paired RGB-D data together with 2D detectors to generate 3D pseudo labels to address the label scarcity issue, as shown in Figure 1 left. But they are still restricted by the small scale of existing paired RGB-D data. Moreover, the from scratch trained 3D detector can hardly inherit from powerful open-vocabulary 2D detector models directly due to the modality difference. We then ask the question, what is the best way to transfer 2D knowledge to 3D for OV-3Det?

Observing that the modality gap prevents a direct knowledge transfer, we propose to leverage a pseudo multi-modal representation to close the gap. On one hand, we can lift a 2D image into a pseudo-3D representation through estimating the depth and camera matrix. On the other hand, we can convert a 3D point cloud into a pseudo-2D representation through rendering. The pseudo RGB image-PC multimodal representation could serve as a common ground for better transferring knowledge from 2D to 3D.

In this paper, we present ImOV3D, which addresses these challenges by employing pseudo-multimodal representation as a unified framework. As shown in Figure 1 right side, In both the training and the inference phase, we construct pseudo-multimodal representation to achieve our goal of training solely with 2D images and better integrating multimodal features to enhance the performance of OV-3Det. Our key idea lies in proper modality conversion. Specifically, the entire pipeline consists of two flows: (1) **Image → Pseudo PC**, by leveraging a large-scale 2D images training set, our method begins by converting images to pseudo point clouds through monocular depth estimation and approximate camera parameter. We automatically generate pseudo 3D labels based on 2D annotations, providing the necessary training data. We also designed a set of revision modules, which significantly improve the quality of the pseudo 3D data through the use of GPT-4 [1]'s size prior and the orientation of the estimated normal map. (2) **Pseudo PC → Pseudo Image**, we learn a point clouds renderer capable of producing natural-looking textured 2D images from pseudo 3D point clouds. This enables ImOV3D to leverage pseudo-multimodal 3D detection even for point cloud-only inputs during inference, transferring 2D rich semantic information and proposals into the 3D space, further enhancing the detector's performance.

Despite being trained solely with the 2D image set, ImOV3D exhibits impressive detection results when directly processing real 3D scans. This is attributed to the high fidelity of the lifted point clouds and the point clouds rendering. Additionally, when a small amount of real 3D data becomes available, even without any 3D annotations, ImOV3D can further narrow the gap between pseudo and real data by fine-tuning on such 3D data, leading to improved detection performance. To validate the effectiveness of ImOV3D, we perform extensive experiments on two benchmark datasets: SUNRGBD [43] and ScanNet [12]. Notably, in scenarios where real 3D training data is unavailable, ImOV3D surpasses previous state-of-the-art open-vocabulary 3D detectors by an mAP@0.25 improvement of at least 7.14% on SUNRGBD and 6.78% on ScanNet. Furthermore, when real 3D training data is accessible, ImOV3D continues to outperform various challenging baselines by a large margin. Thorough ablations are also conducted to validate the efficacy of our designs. In summary, our contributions are three-fold:

- We propose ImOV3D, the first OV-3Det method that can be trained solely with 2D images **without requiring any 3D point clouds or 3D annotations**.

- We introduce a novel **pseudo-multimodal representation** pipeline which converts 2D internet images and corresponding detections into pseudo point clouds, pseudo 3D annotations, and point clouds renderings to support point clouds-based multimodal OV-3Det.

- ImOV3D achieves state-of-the-art performance on two general OV-3Det benchmark datasets across various settings, showcasing its ability to enhance open world 3D understanding despite the lack of 3D data and annotations.

## 2 Related Work

**Open-Vocabulary 2D Object Detection** encompasses two primary series of works: The first [50, 19, 32, 42, 3, 15, 37, 45, 11, 9, 50], which draws upon knowledge from pre-trained Vision-Language models (e.g., CLIP [40]), comprehends the relationships between images and their corresponding textual descriptions, thereby enhancing object recognition and classification. The second series [26, 47, 48, 51, 17, 57, 33, 25, 58] involves the use of extensive training data, specifically text/image pairs, enabling the model to learn a more diverse set of object representations. Detic [58] leverages vocabularies from image classification datasets to train the classification head of object detectors, addressing the issue of insufficient training data and enabling inference on a larger vocabulary set. In the 2D component of our pseudo- multimodal detector, we utilize Detic [58] to predict 2D labels and bounding boxes. These 2D visual information features are then converted and augmented for the 3D point cloud detector, significantly enhancing our model's ability to recognize a broader range of objects.

**Open-Vocabulary Scene Understanding** has recently gained increased attention [36, 41, 54, 21, 22, 24, 14, 10] and plays a critical role in robotics, autonomous driving, *etc*. OpenScene [36] achieves open-world scene understanding without the need for labeled data by densely embedding 3D scene points together with text and image pixels into the CLIP [40] feature space. PLA [14] develops a hierarchical approach to pairing 3D data with text for open-world 3D learning. We focus on OV-3Det, where merely extracting CLIP [40] features is insufficient. We also require the intricate spatial structure of point clouds to enhance detection accuracy and robustness. By integrating both CLIP [40]'s visual knowledge and the detailed geometric information from point clouds, our approach aims to enable the recognition of a broader range of objects beyond the predefined categories.

**Open-Vocabulary 3D Object Detection** in 3D vision is still in its early stages, especially when compared to traditional 3D object detection [38, 39, 34, 27, 6]. OV-3DETIC [29] leverages ImageNet1K [13] to expand the detector's vocabulary set and conducts contrastive learning between images and point clouds modalities for more effective knowledge transfer. OV-3DET [30] generates pseudo 3D annotations for localization using a pre-trained 2D open-vocabulary detector [58]. CoDA [5] leverages 2D and 3D prior information and a cross-modal alignment module to simultaneously learn the localization and classification capabilities. CoDAv2 [7] improves CoDA [5] further by proposing the novel object enrichment strategy and 2D box guidance. FM-OV3D [52] combines multiple foundation models without the need for 3D annotations. However, they are still subject to the influence of the volume of 3D data and still require strict correspondence between RGB-D data. Our method can generate training data for OV-3Det task using only 2D images, without any 3D ground truth data. It can directly achieve state-of-the-art performance when tested on the evaluation set. The designed pseudo-multimodal representation pipeline provides a novel solution for the utilization of both 2D and 3D information.

## 3 Method

### 3.1 Overview

An overview of the proposed open world 3D Object Detection model, **ImOV3D**, is shown in Figure 2. ImOV3D is a point cloud-only model that addresses the challenges of the scarcity of annotated 3D datasets in open-vocabulary 3D Object Detection. To overcome this challenge, ImOV3D uses large-scale 2D datasets to generate pseudo 3D point clouds and annotations. We use a monocular depth estimation model to create metric depth images, which are then converted into pseudo 3D point clouds for both indoor and outdoor scenes. To generate pseudo 3D annotations, we lift 2D bounding boxes into 3D space. To leverage multimodal data, we transform the point clouds into pseudo images using a point cloud renderer. Our training strategy involves a two-stage approach. Firstly, we conduct

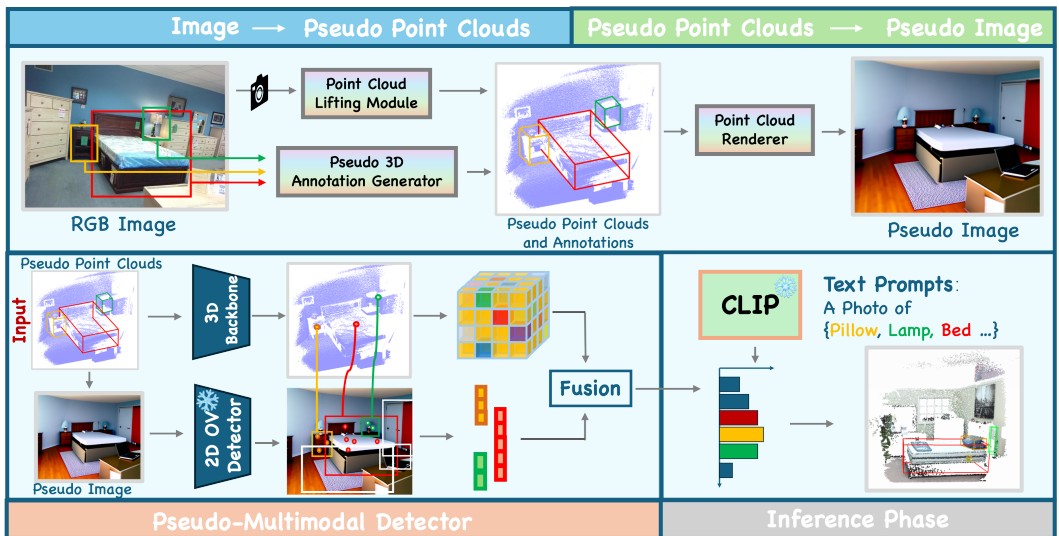

Figure 2: Overview of ImOV3D: Our model takes 2D images as input and puts them into the Pseudo 3D Annotation Generator to produce pseudo annotations. These 2D images are also fed into the Point Cloud Lifting Module to generate pseudo point clouds. Subsequently, using the Point Cloud Renderer, these pseudo point clouds are rendered into pseudo images, which then get processed by a 2D open vocabulary detector to detect 2D proposals and transfer the 2D semantic information to 3D space. Armed with pseudo point clouds, annotations, and pseudo images data, we proceed to train a multimodal 3D detector.

pre-training using pseudo 3D point clouds and corresponding annotations. Subsequently, we initiate an adaptation stage aimed at minimizing the domain discrepancy between 2D and 3D datasets.

## 3.2 Point Cloud Lifting Module

The success of open-vocabulary object detection relies heavily on the availability of large-scale labeled datasets. However, the scarcity of comparable 3D datasets poses a challenge for open world 3D Object Detection. To address this, we bridge 2D images $\mathcal{I}_{\text{2D}} \in \mathbb{R}^{M \times H \times W \times 3}$ (where $M$ is the number of images, and $H$ and $W$ are the height and width, respectively) to 3D detection by generating pseudo 3D point clouds $\mathcal{P}_{\text{pseudo}} \in \mathbb{R}^{M \times N \times 3}$ (where $N$ is the number of points, each with coordinates (x, y, z)).

Utilizing 2D datasets for 3D detection presents difficulties due to the absence of metric depth images and camera parameters. To overcome these obstacles, we use a metric depth estimation model to obtain single-view depth images $\mathcal{D}_{\text{metric}} \in \mathbb{R}^{M \times H \times W}$. Additionally, we employ fixed camera intrinsics $K \in \mathbb{R}^{3 \times 3}$, with the focal length $f$ calculated based on a 55-degree field of view (FOV) and the image dimensions.

However, the absence of camera extrinsics $\mathbf{E} = \{R \mid t\}$ (where $R$ is the rotation matrix and $t$ is the translation vector set to $[0, 0, 0]^\top$) results in the arbitrary orientation of point clouds. To correct this, we use a rotation correction module to ensure the ground plane is horizontal, as shown in Figure 3 (a). First, we estimate the surface normal vector at each pixel using a normal estimation model [2], creating a normal map. From this, we selectively extract the horizontal normal vector $N_i$ at each pixel, defined as $(N_x, N_y, N_z)$. We then compute the normal vector of the horizon surface as $N_{\text{pred}} = Cluster(N_i)$. To align $N_{\text{pred}}$ with the $Z_{axis}$, we calculate the rotation matrix $R$ using the following equation:

$$R = I + K + K^2 \frac{1 - N_{pred} \cdot Z_{axis}}{\|v\|^2} \tag{1}$$

where $I$ is the identity matrix, $v$ is the cross product of $N_{pred}$ and $Z_{axis}$, expressed as $N_{pred} \times Z_{axis}$, $K$ is the skew symmetric matrix constructed from the vector $v$, represented as:

$$K = \begin{bmatrix} 0 & -v_z & v_y \\ v_z & 0 & -v_x \\ -v_y & v_x & 0 \end{bmatrix} \tag{2}$$

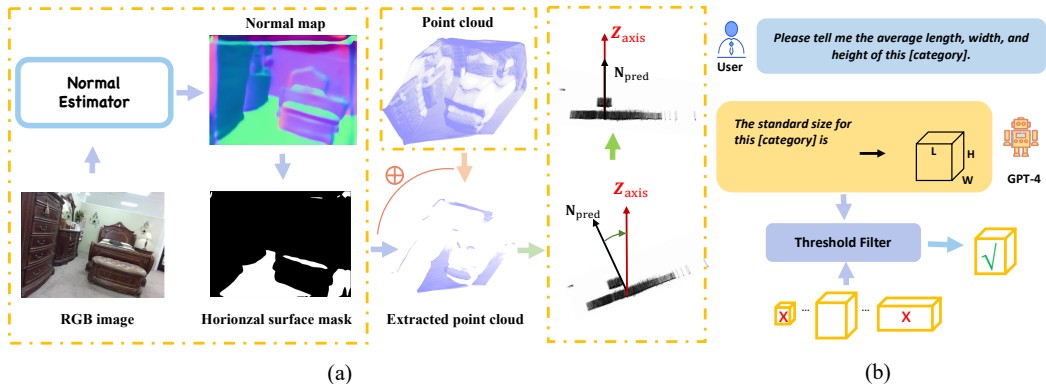

(a)                                   (b)

Figure 3: Illustration of 3D Data Revision Module: **(a)** The rotation correction module involves processing an RGB image through a Normal Estimator to generate a normal map. This map then helps extract a horizontal surface mask for identifying horizontal point clouds, from which normal vectors $N_{pred}$ are obtained. These vectors are aligned with the Z-axis to compute the rotation matrix $R$. **(b)** In the 3D box filtering module, prompts related to object dimensions are first provided to GPT-4 to determine the mean size for each category. This mean size is then used to filter out boxes that do not meet the threshold criteria.

After obtaining the camera intrinsics matrix $K$ and the camera extrinsics matrix $\mathbf{E}$ through the previous steps, depth images $\mathcal{D}_{\text{metric}}$ are converted into point clouds $\mathcal{P}_{\text{pseudo}}$.

### 3.3 Pseudo 3D Annotation Generator

Building upon the vast collection of pseudo 3D point clouds $\mathcal{P}_{\text{pseudo}}$ acquired from 2D datasets, our next step is to generate pseudo 3D bounding boxes $\mathcal{B}_{\text{3Dpseudo}} \in \mathbb{R}^{M \times K \times 7}$(where $K$ is the number of bounding boxes and each box has 7 parameters: center coordinates, dimensions, and orientation).

2D datasets contain rich segmentation information that can be used to generate 3D boxes by lifting. Using the camera intrinsics matrix $K$ and camera extrinsics matrix $\mathbf{E}$ obtained through the Point Cloud Lifting Module, we lift the 2D bounding boxes $\mathcal{B}_{2Dgt} \in \mathbb{R}^{M \times K \times 4}$ from the 2D datasets into 3D space by extracting 3D points that fall within the predicted 2D boxes, generating frustum 3D boxes $\mathcal{B}_{\text{3Dpseudo}}$. The extracted point clouds may contain background points and outliers. To remove these, we employ a clustering [16] algorithm to analyze point clouds. Through the clustering results, we can identify and remove background points and outliers that do not belong to the target objects.

However, these lifted 3D boxes may still contain noise from the depth images $\mathcal{D}_{\text{metric}}$ obtained by monocular depth estimation. To address this issue, we use a 3D box filtering module to filter out inaccurate 3D boxes, as shown in Figure 3 (b). First, we construct a database of median object sizes using GPT-4 [1]. By prompting GPT-4 with "Please tell me the average length, width, and height of this [category], using meters as the unit", we obtain the median dimensions $L_{GPT}, W_{GPT}, H_{GPT}$. Each object in a scene, defined by dimensions $L, W, H$, is compared to these median dimensions using a threshold $T$. An object is preserved if each element of:

$$T < \frac{R}{R_{\text{GPT}}} < \frac{1}{T}, \quad \forall R \in \{L, W, H\} \tag{3}$$

The 3D box filtering module consists of two components: Train Phase Prior Size Filtering and Inference Phase Semantic Size Filtering. The first component filters out boxes that do not match the size criteria before training. The second component removes semantically similar but size-different categories during inference, preventing errors such as misidentifying a book as a bookcase.

### 3.4 Point Cloud Renderer

Point cloud data has inherent limitations, such as the inability of sparse point clouds to capture detailed textures. 2D images can enrich 3D data by providing additional texture information that point clouds lack. To utilize 2D images, we transform point clouds $\mathcal{P}$ into rendered images $\mathcal{I}_{\text{rendered}} \in \mathbb{R}^{M \times H \times W}$.

Integrating rendered images into a 3D detection pipeline is challenging. A naive approach, as mentioned in PointClip [59], is to append raw depth values across the RGB channels, but this fails to apply a mature open-world 2D detector effectively. To leverage multimodal information without

additional inputs beyond 3D point clouds, we develop a point cloud renderer to convert point clouds into detailed pseudo images. This process can also be learned solely from 2D image datasets.

The point cloud renderer has two key modules: The point cloud rendering module converts point clouds $\mathcal{P}$ into rendered images $\mathcal{I}_{\text{rendered}}$, and the color rendering module then processes these images to produce colorized outputs using ControlNet [55]. ControlNet [55] is a method designed to control diffusion models, transforming rendered images $\mathcal{I}_{\text{rendered}}$ into pseudo images $\mathcal{I}_{\text{pseudo}} \in \mathbb{R}^{M \times H \times W \times 3}$.

In the pretraining stage, we use the camera intrinsics $K$ and extrinsics $\mathbf{E}$ from the Point Cloud Lifting Module to render $\mathcal{P}_{\text{pseudo}}$ into rendered images $\mathcal{I}_{\text{rendered}}$. During adaptation and inference, we render ground truth point clouds $\mathcal{P}_{\text{gt}}$ into images using the intrinsics $K$ obtained in the same way. Due to the lack of extrinsics $\mathbf{E}$, the final rendered images $\mathcal{I}_{\text{rendered}}$ are obtained by finding the optimal angle from different horizontal and vertical perspectives to make the images most compact.

In reality, we cannot project a point cloud while guaranteeing that every pixel corresponds to some points. There will be holes and missing areas due to point cloud imperfections or incompatible viewpoint selection. We adjust the camera's position horizontally and vertically to observe point clouds from various angles, removing obscured portions. Finally, we render the point clouds back into images from their original perspective, resulting in partial view rendered images $\mathcal{I}_{\text{partial}} \in \mathbb{R}^{M \times N \times 3}$. The angle range for adjustments is set from -75 to 75 degrees, with a 15-degree interval:

$$\theta_h, \theta_v \in \{-75 + 15k | k = 0, 1, 2, \ldots, 10\}^\circ \tag{4}$$

where $k$ is an integer indicating the stepwise adjustment of the camera's angle.

After generating partial view rendered images $\mathcal{I}_{\text{partial}}$, the next step is to fine-tune ControlNet [55] using these images to obtain pseudo images $\mathcal{I}_{\text{pseudo}}$. Three types of data are prepared for fine-tuning: prompts, targets, and sources. RGB images $\mathcal{I}_{\text{2D}}$ from a 2D dataset serve as the targets, while the partial view rendered images $\mathcal{I}_{\text{partial}}$ are the training sources. Prompts are not used during training.

Finally, we use the pseudo images $\mathcal{I}_{\text{pseudo}}$ and annotations $\mathcal{B}_{\text{2Dgt}}$ from 2D datasets to fine-tune an open-vocabulary 2D detector. Thus, we can use $\mathcal{I}_{\text{pseudo}}$ to obtain corresponding pseudo 2D bounding boxes $\mathcal{B}_{\text{2DTpseudo}} \in \mathbb{R}^{M \times K \times 4}$.

### 3.5 Pseudo Multimodal 3D Object Detector

With an extensive dataset comprising abundant 3D data ($\mathcal{P} + \mathcal{B}_{\text{3Dpseudo}}$) and pseudo images data $\mathcal{I}_{\text{pseudo}}$, our next step is to train a pseudo multimodal 3D detector using a two-stage approach.

**Training Strategy** Our training process includes pretraining and adaptation stages. In the pretraining stage, we train on pseudo 3D point clouds $\mathcal{P}_{\text{pseudo}}$ and annotations $\mathcal{B}_{\text{3Dpseudo}}$, combined with pseudo images $\mathcal{I}_{\text{pseudo}}$. While the pre-trained model performs well for zero-shot detection, a significant domain gap exists between 2D and 3D datasets.

In the adaptation stage, to minimize the domain gap, we follow the same approach as OV-3DET. First, a pre-trained open-vocabulary 2D detector is used to detect objects in the image. Then, these 2D boxes $\mathcal{B}_{\text{2Dpseudo}} \in \mathbb{R}^{M \times K \times 4}$, along with RGBD data, are lifted into 3D space. Through clustering to remove background and outlier points, we obtain precise and compact 3D boxes $\mathcal{B}_{\text{3Dpseudo}}$. Finally, this processed data is used for adaptation. To further explore the benefits of pretrain, we use 3D datasets of varying sizes to test the model's performance under different data availability conditions.

**Loss Function** In this section, we describe loss function used in the pretrain stage. By leveraging $\mathcal{P}_{\text{pseudo}}$ and $\mathcal{B}_{\text{3Dpseudo}}$, a 3D backbone is trained to obtain seed points $\mathcal{K} \in \mathbb{R}^{K \times 3}$, where $K$ represents the number of seeds, along with 3D feature representations $F_{pc} \in \mathbb{R}^{K \times (3+F)}$, with $F$ denoting the feature dimension. Then, seed points are projected back into 2D space via the camera matrix. These seeds that fall within the 2D bounding boxes $\mathcal{B}_{\text{2DTpseudo}}$ retrieve the corresponding 2D cues associated with these boxes and bring them back into 3D space. These lifted 2D cues features are represented as $F_{img} \in \mathbb{R}^{K \times (3+F')}$, where $F'$ represents the feature dimension. Finally, the point cloud features $F_{pc}$ and image features $F_{img}$ are concatenated, forming the joint representation $F_{joint} \in \mathbb{R}^{K \times (3+F+F')}$. In the adaptation stage, $\mathcal{P}_{\text{pseudo}}$ is replaced with $\mathcal{P}_{\text{gt}}$, keeping the workflow consistent with the pretrain stage.

$$\mathcal{L}_{\text{total}} = \mathcal{L}_{\text{loc}} + \sum_i W_i \times \text{CrossEntropy}(\text{Cls-header}(\mathcal{F}_i) \cdot \mathcal{F}_{\text{text}}) \tag{5}$$

where $i$ represents different features, such as pc, img, joint. $W_i$ is the weight corresponding to feature $i$. $\mathcal{L}_{\text{loc}}$ represents the original localization loss function used in ImVoteNet[39]. $\mathcal{F}_{\text{text}}$ denotes the feature extracted by the text encoder in CLIP.

**Implementation Details** Our model is a point cloud-only ImVoteNet [39] +Clip architecture. The monocular depth estimation model used is ZoeDepth[4], jointly trained on both indoor and outdoor scenes. In the pre-training phase, similar to ImVoteNet, we train for 180 epochs with an initial learning rate of 0.001. In the adaptation phase, we train for 100 epochs, reducing the learning rate to 0.0005.

For 2D voting, there are three types of cues: Geometric cues, Texture cues, and Semantic Cues. Unlike ImVoteNet, we retain geometric cues but remove texture cues. For Semantic cues, instead of using a one-hot class vector, we use pre-trained CLIP text encoder features, which is more suitable for an open-vocabulary setting.

## 4 Experiments

In this section, we compare our proposed ImOV3D with other baseline models. Our experimental setup is divided into two main stages: **Pretraining** and **Adaptation**. During the pretraining stage, the training data is pseudo 3D data, referring to pseudo 3D point clouds and their corresponding annotations (3D boxes). During the adaptation stage, we use ground truth point clouds and pseudo labels to minimize the domain gap. All experiments are conducted on two commonly used Object Detection datasets: SUNRGBD [43] and ScanNet [12]. Additionally, we carry out comprehensive ablation studies to validate the effectiveness of our model's components and the data generation pipeline.

### 4.1 Experimental Setup

**2D Images Dataset:** We select the LVIS [20] dataset as our 2D image source for generating pseudo 3D data, utilizing 42,000 images provided in its training set, which spans 1,203 categories with rich and detailed annotations.

**3D Point Clouds Dataset:** We select SUNRGBD [43] and ScanNet [12] as our 3D point clouds datasets for adaptation and testing, SUNRGBD [43] and ScanNet [12] encompass a diverse range of indoor environments and offer comprehensive annotations, including 2D and 3D bounding boxes for objects. We test on 20 common categories in both datasets.

**Evaluation Metrics:** We employ mean Average Precision (mAP) at an IoU threshold of 0.25 as our primary evaluation metric. This metric effectively balances precision and recall in assessing how well our models perform on selected datasets.

Table 1: Results from the Pretraining stage comparison experiments on SUNRGBD and ScanNet, ImOV3D only require point clouds input.

| Stage | Data Type | Method | Input | Training Strategy | SUNRGBD mAP@0.25 | ScanNet mAP@0.25 |
|---|---|---|---|---|---|---|
| **Pre-training** | **Pseudo Data** | OV-VoteNet [38] | Point Cloud | One-Stage | 5.18 | 5.86 |
| | | OV-3DETR [34] | Point Cloud | One-Stage | 5.24 | 5.30 |
| | | OV-3DET [30] | Point Cloud + Image | Two-Stage | 5.47 | 5.69 |
| | | **Ours** | **Point Cloud** | **One-Stage** | **12.61 ↑ 7.14** | **12.64 ↑ 6.78** |

Table 2: Results from the Adaptation stage comparison experiments on SUNRGBD and ScanNet

| Stage | Method | Input | Training Strategy | SUNRGBD mAP@0.25 | ScanNet mAP@0.25 |
|---|---|---|---|---|---|
| **Adap-tation** | OV-3DET [30] | Point Cloud + Image | Two-Stage | 20.46 | 18.02 |
| | CoDA [5] | Point Cloud | One-Stage | — | 19.32 |
| | **Ours** | **Point Cloud** | **One-Stage** | **22.53↑ 2.07** | **21.45↑ 2.13** |

### 4.2 Main Results

**Pretraining:** Due to the absence of existing baseline methods except OV-3DET [30], we utilize CLIP [40] to make previous high-performance 3D detectors such as 3DETR [34] and VoteNet [38]compatible with OV3Det. Specifically, to adapt traditional point cloud detectors for Open Vocabulary detection, we first extract geometric features from point clouds. Then, we integrate

CLIP [40] for classification by converting these features for compatibility with CLIP [40] visual encoder and creating textual prompts for zero-shot classification. Finally, we compare the encoded prompts with the visual features to classify objects beyond the predefined categories. Therefore, these baselines are denoted as OV-VoteNet [38], OV-3DETR [34].

**Adaptation:** To ensure a fair comparison with the current SOTA OV3Det methods, during the adaptation stage, all baselines use OV-3DET [30]'s approach to generating pseudo labels for ground truth point cloud data, which serve as training data for adaptation. In this stage, comparisons are made with CoDA [5] and OV-3DET [30].

### 4.2.1 Pretraining → 3D Training Data Free OV-3Det

As shown in Table 1, training solely with pseudo 3D data generated by our method, ImOV3D improves mAP@0.25 by 7.14% on SUNRGBD and 6.78% on ScanNet over the best baseline. This achievement, made without using any 3D ground truth annotated data, demonstrates the high quality of our generated data and the effectiveness of using extensive 2D datasets to enhance Open World perception. Unlike OV-VoteNet, which lacks 2D image integration, our method's mAP@0.25 outperforms OV-VoteNet by 7.43% and 6.78% on the two datasets, proving the effectiveness of our multimodal approach even with only point cloud inputs. OV-3DET and ImOV3D visualization results are shown in Figure 6(b).

### 4.2.2 Adaptation → 3D Training Data Guided OV-3Det

Table 2 shows original OV-3DET results in the first row. CoDA only compares with OV-3DET on ScanNet. Our experiments indicate that after pretraining with pseudo 3D data, ImOV3D outperforms the best baseline by 2.07% on SUNRGBD and 2.13% on ScanNet in mAP@0.25. This highlights the crucial role of pseudo 3D data in training and its effectiveness as data augmentation.

## 5 Ablation Study

### 5.1 Ablation Study of 3D Data Revision

To validate the effectiveness of enhancing pseudo 3D data quality, we conducted ablation experiments with the Rotation Correction Module and 3D Box Filtering Module. The 3D Box Filtering Module includes Train Phase Prior Size Filtering and Inference Phase Semantic Size Filtering. Table 3 shows the results: the baseline without any modules, adding Train Phase Prior Size Filtering improves mAP@0.25 by 1.65% on SUNRGBD and 1.27% on ScanNet. Adding the Rotation Correction Module improves by 1.3% on SUNRGBD and 1.96% on ScanNet. Combining both modules results in a 2.98% improvement on SUNRGBD and 3.31% on ScanNet. Adding Semantic Size Filtering during inference further increases mAP@0.25 by 4.26% on SUNRGBD and 4.31% on ScanNet. These results highlight the effectiveness of each module in improving data quality and OV3Det accuracy.

Table 3: Results from the ablation study on the Rotation Correction Module and the 3D Box Filtering Module, conducted on SUNRGBD and ScanNet, are presented. The 3D Box Filtering Module is divided into two components: Train Phase Prior Size Filtering and Inference Phase Semantic Size Filtering.

| Stage | Train Phase Prior Size | Rotation Correction | Inference Phase Semantic Size | SUNRGBD mAP@0.25 | ScanNet mAP@0.25 |
|-------|------------------------|---------------------|-------------------------------|------------------|------------------|
| **Pre-training** | ✗ | ✗ | ✗ | 8.35 | 8.33 |
| | ✓ | ✗ | ✗ | 10.00 | 9.60 |
| | ✗ | ✓ | ✗ | 9.65 | 10.29 |
| | ✓ | ✓ | ✗ | 11.33 | 11.64 |
| | ✓ | ✓ | ✓ | **12.61** | **12.64** |

We also discuss the efficiency of GPT-4 [1] in the 3D box filtering module using the SUNRGBD dataset [43]. For comparison, we select the top 10 classes with the most instances in the validation set. The volume ratio for these 10 classes is defined as $\text{Ratio}_V = \frac{L \times W \times H}{L_{\text{GT/GPT}} \times W_{\text{GT/GPT}} \times H_{\text{GT/GPT}}}$. This ratio is an insightful metric for comparing the performance of the GPT-4 powered 3D box filter module

to the ground truth (GT). A ratio close to 1 indicates high precision. We calculate $Ratio_V$ for each instance and use Kernel Density Estimation (KDE) to analyze and plot the distributions of the volume ratios. Results are presented in Figure 6(a).

## 5.2 Ablation Study of Depth and Pseudo Images

To validate the effectiveness of pseudo images generated by ControlNet [55], we compare 2D depth maps from pseudo point clouds with pseudo images, shown in Figure 4. On the SUNRGBD dataset, mAP@0.25 increased from 4.38% to 12.61%, and on the ScanNet dataset, it rose from 4.47% to 12.64% (see Table 4). This shows that rich texture information in 2D images significantly enhances 3D detection performance.

Table 4: The results from different types of 2D rendering images include depth maps and pseudo images.

| Stage | Rendered Images Data Types | SUNRGBD mAP@0.25 | ScanNet mAP@0.25 |
|---|---|---|---|
| Pre-training | Depth Map | 4.38 | 4.47 |
| | Pseudo Images | **12.61** | **12.64** |

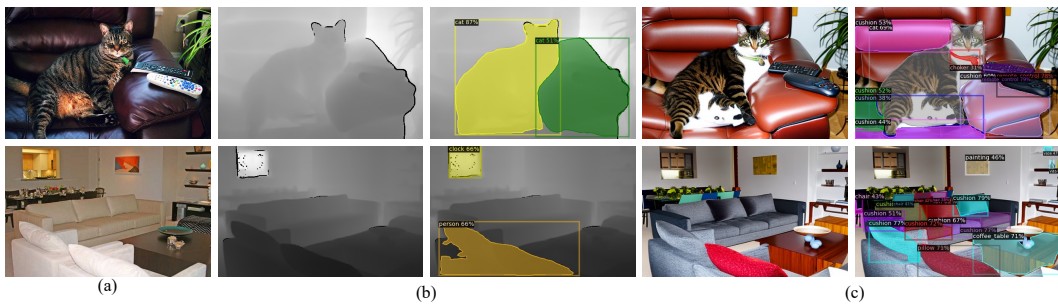

Figure 4: Qualitative results include (a) 2D RGB images, (b) 2D depth maps with 2D OVDetector annotations, and (c) pseudo images with annotations from a fine-tuned 2D detector.

## 5.3 Ablation Study of Data Volume

Our method fine-tunes with limited real ground truth 3D point cloud data and pseudo 3D annotations. Using OV-3DET's code, we train with varying data volumes. With 10% adaptation data, OV-3DET's mAP@0.25 on SUNRGBD drops from 20.46% to 15.24%, while ours drops from 22.53% to 19.24%. On ScanNet, OV-3DET's mAP@0.25 falls from 18.02% to 14.35%, and ours falls from 21.45% to 18.45% (Figure 5(a)(b)). We observed a decrease in performance compared to using the full data set; however, our method was still able to maintain relatively high detection accuracy. This confirmed the robustness of our method and its adaptability to small datasets, enabling effective 3D Object Detection even under constrained data conditions. It also underscores the importance of developing models for OV3Det that are capable of learning from limited data and generalizing to a broader range of scenarios.

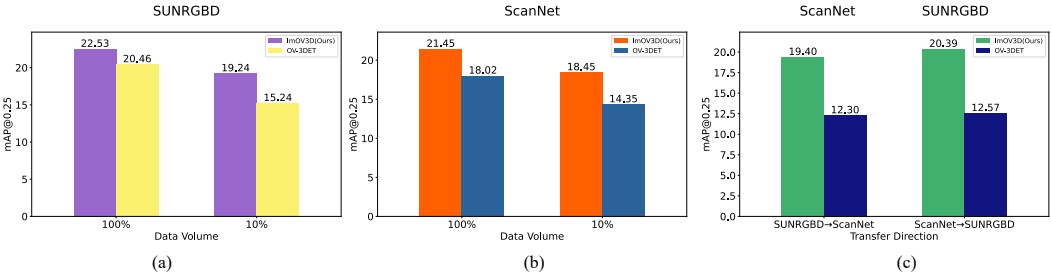

Figure 5: (a) and (b) show data volume ablation results. (c) illustrates transferability ablation results.

## 5.4 Analysis of Transferability

Traditional 3D detectors struggle with transferability due to training and testing class differences. We test ImOV3D on ScanNet and SUN RGB-D on the opposite datasets. The results, shown in Figure

5(c), demonstrate that our model outperforms OV-3DET by 7.1% on SUN RGB-D and 7.82% on ScanNet. ImOV3D demonstrates superior transferability across domains despite the domain gap.

## 5.5 Analysis of Fine-tuned 2D Detector

Table 5: Comparison of fine-tuned 2D detector: Off-the-Shelf vs. Fine-Tuned Detic.

| Pretraining | Adaptation | SUNRGBD mAP@0.25 | ScanNet mAP@0.25 |
|---|---|---|---|
| - | 2D Off-the-shelf + 3D Adaptation | 18.8 | 18.96 |
| Off-the-shelf + 3D Pretraining | 2D Off-the-shelf + 3D Adaptation | 19.67 | 19.25 |
| 2D Pretraining + 3D Pretraining | 2D Adaptation + 3D Adaptation | **22.53** | **21.45** |

To validate the benefits of fine-tuning Detic with pseudo images, we compare the off-the-shelf Detic to the fine-tuned version. The fine-tuned Detic shows clear advantages in handling pseudo images. On the SUNRGBD dataset, the mAP@0.25 increases from 19.67% to 22.53%, and on the ScanNet dataset, it rises from 19.25% to 21.45% (see Table 5). These experiments were conducted under the adaptation setting, illustrating the model's ability to learn from and improve detection capabilities with not entirely real data. This not only confirms the efficacy of textured image but also highlights the importance of fine-tuning models to enhance their adaptability and accuracy.

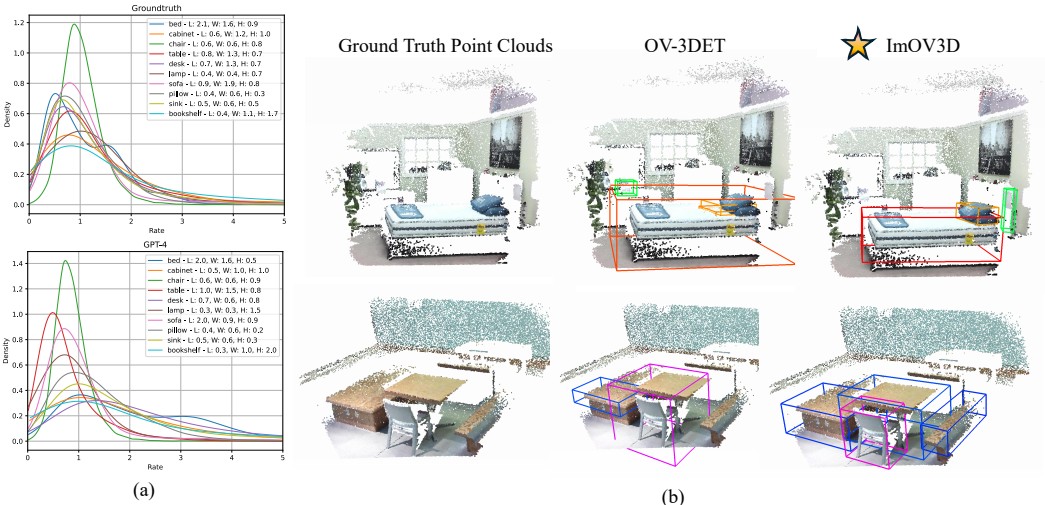

(a)                                    (b)

Figure 6: (a) KDE plots of volume ratios ($Ratio_V$) for top 10 classes in SUNRGBD validation set. (b) Visualization comparison of OV-3DET with ours in SUNRGBD.

## 6 Conclusion and Limitation

In conclusion, this paper introduce ImOV3D, a novel framework that tackles the scarcity of annotated 3D data in OV-3Det by harnessing the extensive availability of 2D images. The framework's key innovation lies in its flexible modality conversion, which integrates 2D annotations into 3D space, thereby minimizing the domain gap between training and testing data. Empirical results on two common datasets confirm ImOV3D's superiority over existing methods, even without ground truth 3D training data, and its significant performance boost with the addition of minimal real 3D data for fine-tuning. Our method's success showcases the potential of leveraging 2D images for enhancing 3D object detection, opening new avenues for future research in pseudo-multimodal data generation and its application in 3D detection methodologies.

**Limitation:** Although our method has demonstrated the potential of 2D images in OV-3Det tasks, especially with the proposed pseudo multimodal representation, we need dense point clouds here to ensure that the rendered images can help improve performance. In the future, we will explore more generalized strategies.

## 7 Acknowledge

We would like to express our gratitude to Yuanchen Ju, Wenhao Chai, Macheng Shen, and Yang Cao for their insightful discussions and contributions.

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

# Supplementary Materials

## I. Supplementary Method Details

This part supplements the methods mentioned in the main paper with detailed implementations and theoretical underpinnings.

**Partial-View Removal:** To obtain a point cloud from a 2D image, we first use a monocular depth estimation model ZoeDepth [4] to generate a depth map for the image, as shown in Fig 4 (A). Inspired by [56], utilizing partial-view depth images for data augmentation serves to bolster the robustness and generalization capabilities of machine learning models in computer vision tasks. By simulating realistic occlusions and offering a diverse array of viewpoints, this approach prepares models to handle real-world variability and complexity more effectively, improving their performance across a wide range of scenarios.

In this process, we start by defining the point cloud and the camera viewpoints. The point cloud is a collection of points, which is formally represented as:

$$P = \{p_1, p_2, \ldots, p_n\} \tag{6}$$

where each point $p_i$ is a part of the 3D object or scene we are interested in.

We consider two distinct viewpoints, A and B, characterized by their camera parameters $\theta_A$ and $\theta_B$, respectively. These parameters include the position and orientation of the camera that define each viewpoint.

Initially, we observe the point cloud from viewpoint A, resulting in a subset of points observed from this specific viewpoint, taking into account potential occlusions and the camera's field of view. This subset is denoted as:

$$P_A = \text{view}(P, \theta_A) \tag{7}$$

Next, we change our perspective to viewpoint B and observe the point cloud again, which gives us another subset of points, denoted as:

$$P_B = \text{view}(P, \theta_B) \tag{8}$$

The key step in this process involves identifying the overlapping points that are visible from both viewpoints A and B. This overlapping set is represented as:

$$P_{AB} = P_A \cap P_B \tag{9}$$

After identifying the overlapping points, we proceed to remove these points from the subset observed from viewpoint A. The final set of points, after removing the overlapping ones, is represented as:

$$P_{A'} = P_A - P_{AB} \tag{10}$$

Finally, this processed point cloud is rendered back into a depth image from viewpoint A. The resulting depth image, which includes holes where the overlapping points were removed, as shown in Fig 7 (B), illustrating the effect of occlusions, is obtained through the rendering process, expressed as:

$$I_{A'} = \text{render}(P_{A'}, \theta_A) \tag{11}$$

**Normal Estimator:** In this paper, we employ Normal Estimator [2] for surface normal estimation.

The process of normal estimation begins with the calculation of the surface normal vector at each pixel, a crucial step for understanding the orientation of surfaces in a scene. This calculation is conducted using a normal estimation model that utilizes either depth information or the 3D coordinates of each point in the scene. Mathematically, the surface normal vector at a point is defined as a 3D vector $\vec{N} = (N_x, N_y, N_z)$, where $\vec{N}$ represents the direction perpendicular to the surface at that pixel, as shown in the equation

$$\vec{N} = (N_x, N_y, N_z). \tag{12}$$

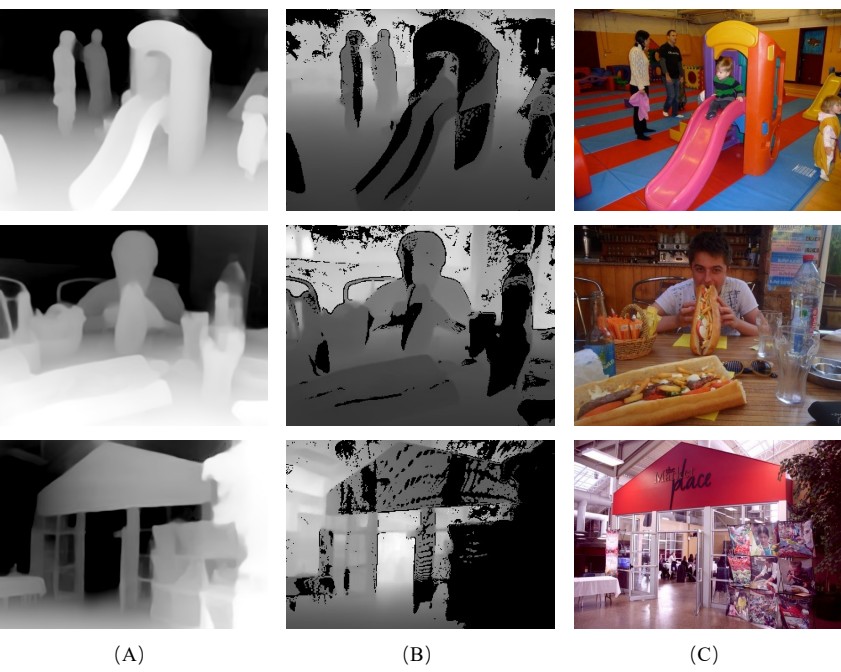

|     (A)     |     (B)     |     (C)     |

Figure 7: Illustration of (A) Depth Images, (B) Rendered Point Cloud Images after Partial-View Removal, and (C) Ground Truth 2D Images

Following the estimation of normal vectors across the scene, the next phase involves the extraction of horizontal normal vectors $\vec{N_i}$ at each pixel. This selection process focuses on vectors that align with the horizontal plane of the scene, essential for applications such as ground plane detection or horizon line estimation. Once horizontal normal vectors are identified, they undergo a clustering process to find a consensus on the predominant horizontal surface orientation within the scene. This process yields $\vec{N_{pred}}$, a vector representing the normal to the horizon surface, encapsulated by the equation

$$N_{pred} = \text{Clustering}(\vec{N_i}). \tag{13}$$

The final step in the normal estimation process is the alignment of $\vec{N_{pred}}$ with the Z-axis of the coordinate system to ensure a standardized representation of orientation. Achieving this alignment involves calculating a rotation matrix $R$, which, when applied to $\vec{N_{pred}}$, orients the vector parallel to the Z-axis. This is crucial for maintaining consistent orientation representation across different scenes and supports applications requiring a fixed reference frame. The alignment can be mathematically represented as

$$N_{aligned} = R \cdot \vec{N_{pred}}, \tag{14}$$

where $\vec{N_{aligned}}$ is the resulting vector aligned with the Z-axis, ensuring a uniform approach to interpreting surface orientations in various applications.

**3D Box Filtering Module:** Similar to [53], this module's goal is to improve annotation quality by converting 2D bounding boxes into their 3D counterparts for objects within indoor scenes. This conversion process is pivotal for achieving a more accurate spatial representation of objects. However, the transition from 2D to 3D can introduce discrepancies between the generated 3D bounding boxes and the objects' actual dimensions. To mitigate this, we employ Large Language Models (LLMs), such as GPT-4 [1], to obtain size priors for various object categories encountered in indoor environments. GPT-4[1]'s capability to provide reliable size estimates ensures that the adjusted 3D bounding boxes more closely align with the real-world sizes of objects, thereby enhancing the precision of annotations in indoor scene analysis.

## II. Unexpanded Category AP Analysis

This section provides a detailed discussion on the 20 categories within the SUNRGBD[43] and ScanNet [12] datasets that were not fully explored in the main paper.

Table 6 and Table 7 corresponds to Table 1 in the main paper.

Table 8 and Table 9 corresponds to Table 2 in the main paper.

| Methods | **Mean** | toilet | bed | chair | sofa | dresser | table | cabinet | bookshelf | pillow | sink |
|---|---|---|---|---|---|---|---|---|---|---|---|
| OV-VoteNet [38] | 5.86 | 17.36 | 25.83 | 0.54 | 28.85 | 0.02 | 0.94 | 0.19 | 0.0027 | 2.75 | 5.32 |
| OV-3DET [30] | 5.69 | 23.44 | 22.39 | 7.07 | 24.15 | 0.98 | 0.21 | 0.12 | 0.03 | 0.18 | 1.35 |
| OV-3DETR [34] | 5.3 | 25.34 | 10.12 | 1.11 | 28.2 | 0.001 | 0.05 | 0.029 | 0.0001 | 3.05 | 2.34 |
| **ImOV3D** (Ours) | **12.64** | 55.86 | 28.82 | 11.72 | 28.16 | 0.453 | 7.44 | 1.23 | 0.05 | 4.42 | 29.12 |

| Methods | | bathtub | refrigerator | desk | nightstand | counter | door | curtain | box | lamp | bag |
|---|---|---|---|---|---|---|---|---|---|---|---|
| OV-VoteNet [38] | | 18.71 | 7.63 | 2.2 | 0.0059 | 0.04 | 0.29 | 1.16 | 0.02 | 5.28 | 0.03 |
| OV-3DET [30] | | 16.98 | 3.1 | 0.1 | 0.0012 | 0.12 | 0.12 | 4.15 | 7.2 | 2.1 | 0.1 |
| OV-3DETR [34] | | 24.68 | 4.24 | 1.6 | 0.0034 | 0.02 | 0.47 | 1.45 | 0.016 | 2.23 | 1.01 |
| **ImOV3D** (Ours) | | 43.61 | 14.90 | 13.05 | 0.09 | 0.17 | 1.77 | 0.53 | 0.56 | 10.34 | 0.50 |

Table 6: Comparison of Average Precision (AP) Scores for 20 Specific Categories on the ScanNet[12] Dataset During the Pretraining Stage. 'Mean' represents the average value of all the 20 categories.

| Methods | **Mean** | toilet | bed | chair | bathtub | sofa | dresser | scanner | fridge | lamp | desk |
|---|---|---|---|---|---|---|---|---|---|---|---|
| OV-VoteNet [38] | 5.18 | 25.16 | 20.63 | 1.59 | 14.43 | 24.98 | 0.0021 | 0.001 | 0.19 | 1.69 | 0.52 |
| OV-3DET [30] | 5.47 | 21.92 | 16.94 | 11.18 | 12.74 | 16.81 | 1.674 | 0.89 | 4.12 | 5.60 | 0.88 |
| OV-3DETR [34] | 5.24 | 33.50 | 8.47 | 2.61 | 26.54 | 17.05 | 0.0003 | 0.0001 | 2.29 | 0.73 | 0.05 |
| **ImOV3D** (Ours) | **12.61** | 53.45 | 21.61 | 13.65 | 49.00 | 25.20 | 0.14 | 0.02 | 4.81 | 8.49 | 3.23 |

| Methods | | table | stand | cabinet | counter | bin | bookshelf | pillow | microwave | sink | stool |
|---|---|---|---|---|---|---|---|---|---|---|---|
| OV-VoteNet [38] | | 0.5 | 0.0004 | 0.0057 | 0.0007 | 0.0007 | 0.0008 | 8.66 | 0.01 | 5.1 | 0.11 |
| OV-3DET [30] | | 1.08 | 0.12 | 0.89 | 0.93 | 0.0003 | 1.47 | 7.43 | 3.54 | 0.13 | 1.01 |
| OV-3DETR [34] | | 0.05 | 0.0001 | 0.0004 | 0.0001 | 0.0001 | 0.0002 | 9.51 | 0.0001 | 2.47 | 1.63 |
| **ImOV3D** (Ours) | | 6.17 | 0.0011 | 0.0354 | 0.0022 | 12.22 | 0.0048 | 14.20 | 11.98 | 20.63 | 7.39 |

Table 7: Comparison of Average Precision (AP) Scores for 20 Specific Categories on the SUNRGBD[43] Dataset During the Pretraining Stage. 'Mean' represents the average value of all the 20 categories.

| Methods | **Mean** | toilet | bed | chair | sofa | dresser | table | cabinet | bookshelf | pillow | sink |
|---|---|---|---|---|---|---|---|---|---|---|---|
| OV-3DET [30] | 18.02 | 57.29 | 42.26 | 27.06 | 31.5 | 8.21 | 14.17 | 2.98 | 5.56 | 23 | 31.6 |
| CoDA [5] | 19.32 | 68.09 | 44.04 | 28.72 | 44.57 | 3.41 | 20.23 | 5.32 | 0.03 | 27.95 | 45.26 |
| **ImOV3D** (Ours) | **21.45** | 79.23 | 52.07 | 29.25 | 60.20 | 0.47 | 21.67 | 2.21 | 1.37 | 23.50 | 42.02 |

| Methods | | bathtub | refrigerator | desk | nightstand | counter | door | curtain | box | lamp | bag |
|---|---|---|---|---|---|---|---|---|---|---|---|
| OV-3DET [30] | | 56.28 | 10.99 | 19.72 | 0.77 | 0.31 | 9.59 | 10.53 | 3.78 | 2.11 | 2.71 |
| CoDA [5] | | 50.51 | 6.55 | 12.42 | 15.15 | 0.68 | 7.95 | 0.01 | 2.94 | 0.51 | 2.02 |
| **ImOV3D** (Ours) | | 51.39 | 25.65 | 30.11 | 0.39 | 0.66 | 1.05 | 0.07 | 2.94 | 2.70 | 2.13 |

Table 8: Comparison of Average Precision (AP) Scores for 20 Specific Categories on the ScanNet[12] Dataset During the Adaptation Stage. 'Mean' represents the average value of all the 20 categories.

| Methods | **Mean** | toilet | bed | chair | bathtub | sofa | dresser | scanner | fridge | lamp | desk |
|---|---|---|---|---|---|---|---|---|---|---|---|
| OV-3DET [30] | 20.46 | 72.64 | 66.13 | 34.8 | 44.74 | 42.10 | 11.52 | 0.29 | 12.57 | 14.64 | 11.21 |
| **ImOV3D** (Ours) | **22.53** | 76.70 | 65.36 | 32.21 | 55.77 | 50.01 | 0.20 | 2.94 | 13.77 | 26.20 | 11.54 |

| Methods | | table | stand | cabinet | counter | bin | bookshelf | pillow | microwave | sink | stool |
|---|---|---|---|---|---|---|---|---|---|---|---|
| OV-3DET [30] | | 23.31 | 2.75 | 3.4 | 0.75 | 23.52 | 9.83 | 10.27 | 1.98 | 18.57 | 4.1 |
| **ImOV3D** (Ours) | | 16.68 | 0.08 | 0.47 | 0.03 | 29.21 | 0.10 | 15.71 | 20.74 | 30.21 | 2.81 |

Table 9: Comparison of Average Precision (AP) Scores for 20 Specific Categories on the SUNRGBD[43] Dataset During the Pretraining Stage. 'Mean' represents the average value of all the 20 categories.

## III. Evaluating the efficiency of GPT-4 in the 3D box filtering module

In this section, we will discuss the efficiency of GPT-4 [1] in the 3D box filtering module. In order to have a good comparison, we choose the SUNRGBD [43] as a visualizing dataset. To clearly demonstrate the results, we select the top 10 classes with the most instances in the validation set as our visualization targets.

The volume ratio for these 10 classes is defined as $\text{Ratio}_V = \frac{L \times W \times H}{(L_{\text{GT/GPT}} \times W_{\text{GT/GPT}} \times H_{\text{GT/GPT}})}$. The volume ratio provides an insightful metric to compare the performance of the 3D box filter module powered by GPT-4 [1] with the ground truth (GT). A ratio close to 1 indicates that the volume of the predicted box is very similar to the true value, indicating high precision. By calculating the volume ratio $\text{Ratio}_V$ for each instance, we gather statistical data and use Kernel Density Estimation (KDE) to analyze and plot the distributions of the volume ratios. Results are presented in Figure 8.

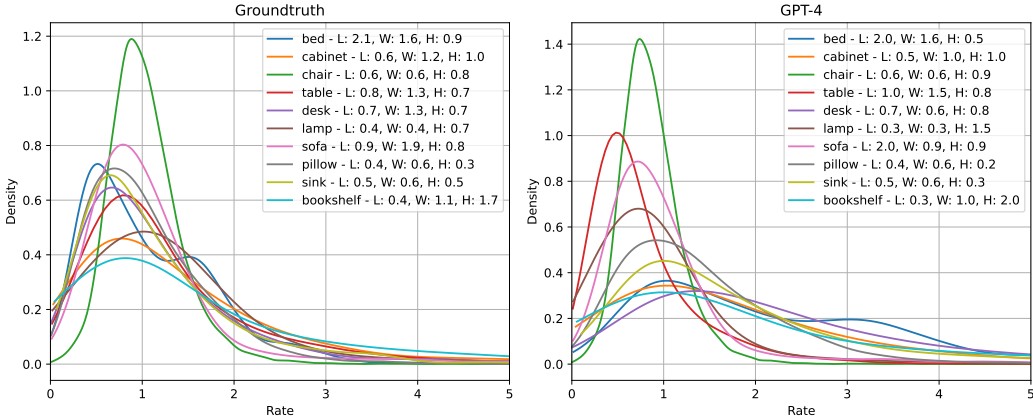

Figure 8: Comparison of Volume Ratios: This image shows the efficiency of GPT-4 [1] using the SUNRGBD dataset by comparing volume ratios of the top 10 classes using Kernel Density Estimation (KDE).

Figure 8 fully demonstrates the effectiveness of the meansize database constructed by GPT-4 [1]. Subsequently, we used the constructed meansize in the 3D box filtering module to filter $\mathcal{B}_{3Dpseudo}$, with the results shown in Figure 9.

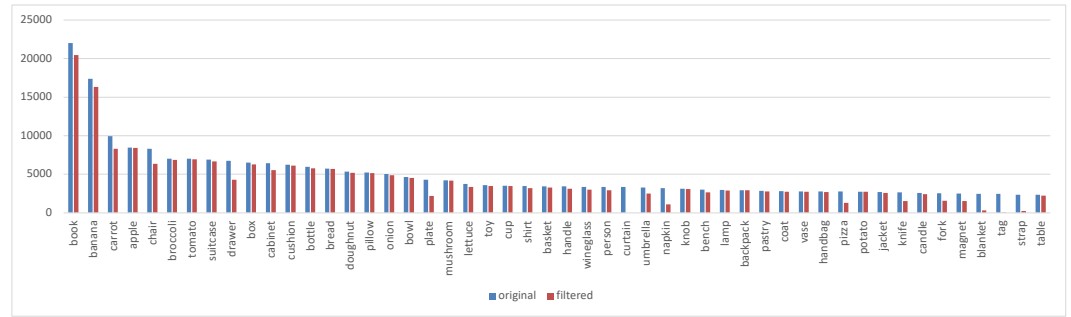

Figure 9: Results of Using the Meansize Database in the 3D Box Filtering Module: This figure shows the outcome of applying the meansize database constructed by GPT-4 [1] in the 3D box filtering module to filter $\mathcal{B}_{3Dpseudo}$ with a threshold $T = 0.1$. The figure presents the number of 3D box of the top 50 classes with the most instances. The results highlight the efficiency and accuracy of the GPT-constructed meansize in improving the performance of the filtering process.

# IV. Detailed Experimental Implementation

## IV.A Fine-tuning Model Details (Detic [58] and ControlNet [55])

This subsection delves into the model adjustment and parameter selection process during the fine-tuning phase.

**Detic [58]:** To finetune Detic [58] for our specific application, we first need to clarify the necessity for such an adjustment. The off-the-shelf Detic model is designed and optimized for ground truth images. However, our requirement for Detic [58] significantly diverges because it is to be utilized within the 2D OV Detector component of our ImOV3D project. In this component, the images we use are not ground truth images but pseudo images generated through ControlNet [55]. Such images differ visually from ground truth images, which could potentially affect the detection performance of the Detic[58] model.

Given this, we decided to finetune Detic [58] to better suit our specific scenario. The finetuning process involves replacing the annotations based on ground truth images, which the Detic [58] model originally uses, with annotations from a 2D dataset that are accurate for our ControlNet-generated pseudo images. This replacement is aimed at enabling Detic [58] to more accurately recognize and process objects within these generated images, considering they differ in quality and characteristics from the real images used in the model's initial training.

Several key steps are involved in this finetuning process. First, we need to collect and prepare annotated data from the 2D dataset that applies to our pseudo images. Then, we use these annotations in the training process of the Detic [58] model to adapt it to our unique image characteristics. Through this process, we ensure that the Detic [58] model maintains efficiency and accuracy when dealing with images generated through ControlNet [55], thereby maximizing its efficacy in our ImOV3D project.

Finetuning the Detic [58] model not only improves its performance on specific images but also further validates the feasibility and flexibility of our approach. This process illustrates that by meticulously adjusting and optimizing model parameters in the face of varying application scenarios, we can effectively enhance the model's performance and adaptability.

**ControlNet [55]:** The objective of finetuning ControlNet [55] is to add color to the rendered point cloud images. Originally, point cloud data lacks color information, meaning the depth images projected from these point clouds are also devoid of color. However, in many applications including but not limited to 3D reconstruction, augmented reality (AR), and virtual reality (VR), rich color and texture information are crucial for enhancing the visual quality of the final images. This necessitates the finetuning of ControlNet [55].

The finetuning process requires two types of data: original images and target images. In our scenario, the original images are depth images obtained by rendering point cloud after partial-view removal, as shown in Fig 7 (B). While these depth images accurately represent the geometric information of the scenes, they lack color information. The target images, on the other hand, are ground truth 2D images corresponding to the point cloud data, as shown in Fig 7 (C), containing the rich color and texture information we aim to incorporate.

During the finetuning of ControlNet [55], we do not use any text prompts but instead adhere to the official parameter settings recommended by ControlNet [55]. We train the model for 180 epochs, through which the model learns how to effectively map the color information from the target images onto the corresponding depth images. This process not only enhances the visual appearance of the rendered images, making them more realistic and rich, but also improves the model's capability to handle color and texture variations across different scenes.

Through such finetuning, ControlNet [55] can more accurately colorize depth images rendered from point clouds, preserving the geometric details from the original point cloud data while also integrating real-world color and texture characteristics. Such improvements are extremely beneficial for subsequent 3D vision applications, such as scene understanding and interaction in 3D. In summary, the process of finetuning ControlNet [55] is a meticulous attempt to adjust and optimize model performance, aiming to achieve better results in handling specific tasks.

**IV.B Main Experiment and Baseline Experiment Details**

It outlines the setup and execution process of the main experiment, including comparisons with baseline experiments.

We detail and present four sets of key pretraining parameters. Table 10 showcases the training configuration for the ImOV3D method, an improved model we propose. Table 11 provides a detailed description of the training parameters for OV-VoteNet[38], serving as another baseline model. Table 12 focuses on OV-3DETR [34], presenting its specific training settings. Finally, Table 13 outlines the training parameter details for the OV-3DET [30] model. These four tables aim to clearly display the specific parameter settings of each method during the pretraining stage, offering a direct perspective for comparison to the readers.

Table 14 presents the parameter configurations of our experiments during the adaptation stage.

Table 10: Key Parameters for ImOV3D Pretraining Configuration

| Parameter | Value | Description |
|---|---|---|
| -tower_weights | 0.3, 0.3, 0.4 | Fusion weights for different input modalities |
| -batch_size | 12 | Number of samples per batch |
| -learning_rate | 0.001 | Initial rate for model training |
| -weight_decay | 0 | L2 regularization coefficient |
| -max_epoch | 180 | Total number of pretraining training epochs |
| -lr_decay_steps | 80, 120, 160 | Epochs where learning rate decays |
| -lr_decay_rates | 0.1, 0.1, 0.1 | Learning rate decay factors |

Table 11: Key Parameters for OV-VoteNet Pretraining Configuration

| Parameter | Value | Description |
|---|---|---|
| -pc_only_weight | 1 | Point clouds only input |
| -batch_size | 12 | Number of samples per batch |
| -learning_rate | 0.001 | Initial rate for model training |
| -weight_decay | 0 | L2 regularization coefficient |
| -max_epoch | 180 | Total number of pretraining training epochs |
| -lr_decay_steps | 80, 120, 160 | Epochs where learning rate decays |
| -lr_decay_rates | 0.1, 0.1, 0.1 | Learning rate decay factors |

Table 12: Key Parameters for OV-3DETR Pretraining Configuration

| Parameter | Value | Description |
|---|---|---|
| -max_epoch | 180 | Training duration in epochs. |
| -base_lr | 7e-4 | Initial learning rate. |
| -batchsize_per_gpu | 12 | Samples per GPU. |
| -weight_decay | 0.1 | L2 regularization. |
| -warm_lr | 1e-6 | Warm-up learning rate. |
| -warm_lr_epochs | 9 | Duration of warm-up phase. |
| -final_lr | 1e-6 | Learning rate for final phase. |

Table 13: Key Parameters for OV-3DET Pretraining Configuration

| Parameter | Local Phase | DTCC Phase |
|---|---|---|
| -max_epoch | 200 | 50 |
| -nqueries | 128 | 128 |
| -base_lr | 4e-4 | 1e-4 |
| -warm_lr_epochs | - | 1 |
| -batchsize_per_gpu | 12 | 12 |
| -final_lr | 1e-5 | 1e-5 |

Table 14: Key Parameters for ImOV3D Adaptation Configuration

| Parameter | Value | Description |
|---|---|---|
| -tower_weights | 0.3, 0.3, 0.4 | Fusion weights for different input modalities |
| -batch_size | 12 | Number of samples per batch |
| -learning_rate | 0.0005 | Initial rate for model training |
| -weight_decay | 0 | L2 regularization coefficient |
| -max_epoch | 100 | Total number of adaptation training epochs |
| -lr_decay_steps | 40, 80 | Epochs where learning rate decays |
| -lr_decay_rates | 0.1, 0.1 | Learning rate decay factors |

## V. Visualization Comparison during Pretraining Stage

The visualization colors have specific meanings, and we're providing ImOV3D the prompt: "*Please help me locate the {category}*". In the image, a bed is represented by red lines, a lamp by light green, a chair by pink, a table by blue, a sofa by purple, a pillow by orange, and a cabinet by dark green.

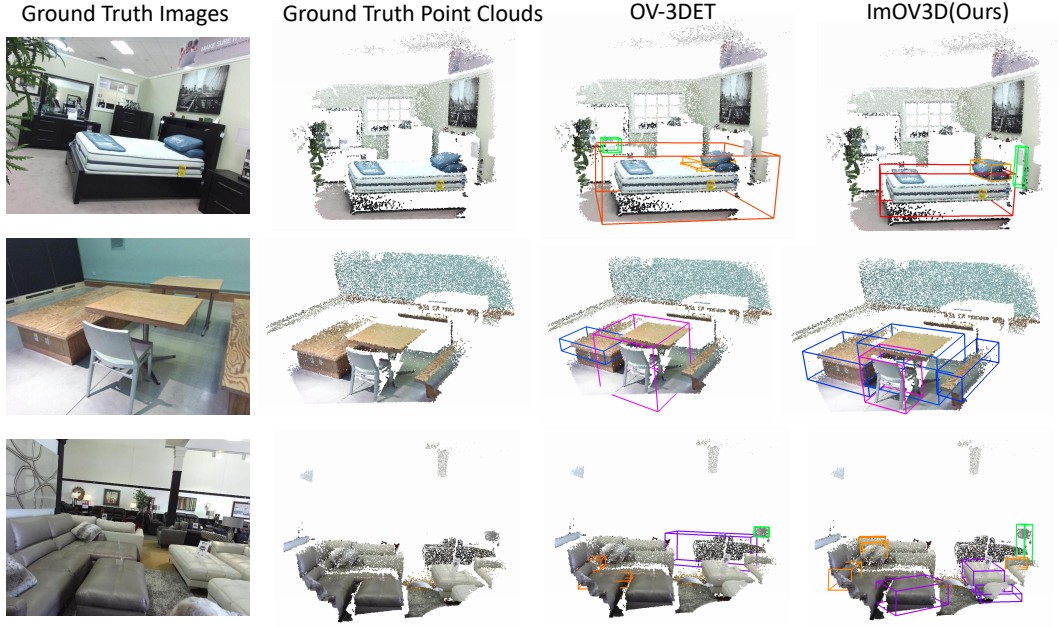

Figure 10: During the pretraining stage, the visual comparison tested on the SUNRGBD[43] dataset includes four columns: the first column shows the ground truth images, the second column displays the ground truth point clouds, the third column presents the detection results by OV-3DET [30], and the fourth column reveals the detection outcomes by ImOV3D.

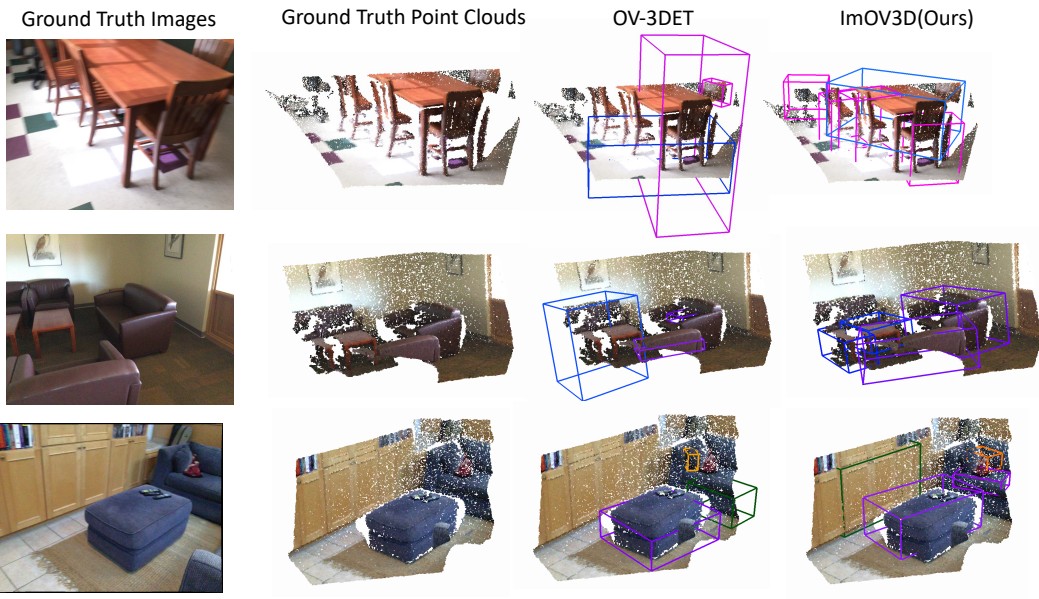

Figure 11: During the pretraining stage, the visual comparison tested on the ScanNet [12] dataset includes four columns: the first column shows the ground truth images, the second column displays the ground truth point clouds, the third column presents the detection results by OV-3DET[30], and the fourth column reveals the detection outcomes by ImOV3D .

## VI. Explanation of the formula for the Rotation Correction Module

In this section, we explore Equation 1 in depth, utilizing Rodrigues' rotation formula. It calculates the rotation matrix $R \in SO(3)$ for a rotation by $\theta$ (less than $180°$) around the axis defined by the unit vector $\hat{n} = (n_x, n_y, n_z)$. The Rodrigues' rotation formula matrix is defined as:

$$R = I + (\sin \theta)N + (1 - \cos \theta)N^2$$

Where $I$ is the identity matrix, and the skew-symmetric matrix $N$ for the unit vector $\hat{n}$ is constructed as:

$$N = \begin{bmatrix} 0 & -n_z & n_y \\ n_z & 0 & -n_x \\ -n_y & n_x & 0 \end{bmatrix}$$

To align unit vector $N_{\text{pred}}$ with unit vector $Z_{\text{pred}}$, we start from the definitions of the inner product and cross product:

$$v = N_{\text{pred}} \times Z_{\text{pred}}$$

$$N_{\text{pred}} \cdot Z_{\text{pred}} = \cos \theta, \quad |N_{\text{pred}} \times Z_{\text{pred}}| = \sin \theta$$

Thus, we have:

$$v = N_{\text{pred}} \times Z_{\text{pred}} \quad \Rightarrow \quad v = \sin \theta \, \hat{n}$$

Accordingly, define:

$$K \overset{\text{def}}{=} (\sin \theta)N = \begin{bmatrix} 0 & -n_z & n_y \\ n_z & 0 & -n_x \\ -n_y & n_x & 0 \end{bmatrix} = \begin{bmatrix} 0 & -v_z & v_y \\ v_z & 0 & -v_x \\ -v_y & v_x & 0 \end{bmatrix}$$

Based on Rodrigues' rotation formula, we have:

$$R = I + K + \frac{1 - \cos \theta}{\sin^2 \theta} K^2$$

$$= I + K + K^2 \frac{1 - N_{\text{pred}} \cdot Z_{\text{pred}}}{\|v\|^2}$$

