# OpenReview forum: "ImOV3D: Learning Open Vocabulary Point  Clouds 3D Object Detection from Only 2D Images"
_NeurIPS.cc/2024/Conference — NeurIPS 2024 poster_

### Official Review · Reviewer_m5vv · 2024-07-12

**Soundness:** 3
**Presentation:** 3
**Contribution:** 2
**Rating:** 5
**Confidence:** 3

**Summary:**

This paper addresses the challenge in Open-vocabulary 3D object detection (OV-3Det), specifically the modality gap between training images and testing point clouds, which hinders effective integration of 2D knowledge into OV-3Det. The main contribution of the paper is a novel method to generate pseudo multimodal data, enabling the training of a 3D encoder to extract 3D features that are coupled with a frozen 2D detector for 3D detection.

**Strengths:**

The provided pseudo-multimodal data generation method used to train the 3D detector, when coupled with the 2D detector, achieves a notable improvement in object detection on the SUNRGBD and ScanNet datasets.

**Weaknesses:**

The framework in Fig. 2 is confusing. The authors claim that their model's input is only point clouds, but the framework suggests that both images and point clouds are needed for inference. Additionally, the generalization ability should be discussed, particularly regarding performance on outdoor datasets.

**Questions:**

What about the generalization ability? For example, how does the model perform on outdoor datasets?

The experiment needs more details. During the inference stage on the ScanNet dataset, do you use only the point cloud information without using the image information?

**Limitations:**

The authors have adequately addressed the limitations.

---

> ### Author Rebuttal · Authors · 2024-08-07
>
> **Q1**：\
> 【***Pre-training Stage***】While our method, ImOV3D, has been primarily designed and evaluated for indoor scenes, we recognize the importance of assessing its performance on real-world 3D data beyond the benchmark datasets mentioned in our paper. To address this, we conducted experiments on the KITTI dataset, which consists of outdoor scenes, to evaluate how well our method generalizes to different datasets.
> ##### Table 1: Results from the Pre-training Stage Comparison of Open Vocabulary 3D Object Detection Performance on KITTI Moderate Difficulty Level
>
> | **Method**   | **Car** | **Pedestrian** | **Cyclist** | **mAP\@0.25** |
> |--------------|---------|----------------|-------------|--------------|
> OV-VoteNet | 11.32 | 8.64 | 8.08 | 9.35 |
> OV-3DETR | 13.68 | 12.08 | 8.56 | 11.44 |
> OV-3DET | 12.92 | 6.24 | 8.52 | 9.22 |
> ImOV3D | 20.42 | 11.24 | 14.96 | 15.54 |
>
> During pre-training stage, the results indicate that our method, while showing strong performance on indoor datasets, also exhibits a reasonable level of robustness and adaptability when confronted with outdoor scenes. Compared to other baselines, our mAP\@0.25 is 4.1% higher. The pretraining images encompass both indoor and outdoor scenes, hence allowing us to naturally generalize to outdoor scenes. \
> 【***Adaptation Stage***】We fine-tuned our model using real point cloud data, and the results show that our method outperforms OV3DET by 5.5% in the mAP\@0.25, indicating a significant enhancement in performance after fine-tuning (compared to the data in Table 1). However, the relatively limited categories in outdoor scenes restrict the full potential of the open-vocabulary detector. Moreover, the differences between indoor and outdoor data also affect the outcomes. More refined adjustments and designs tailored for outdoor scenarios, our results could further improvement.
> ##### Table 2: Results from the Adaptation Stage Comparison of  Open Vocabulary 3D Object Detection Performance on KITTI Moderate Difficulty Level
>
> | **Method**   | **Car** | **Pedestrian** | **Cyclist** | **mAP\@0.25** |
> |--------------|---------|----------------|-------------|--------------|
> OV-3DET | 42.65    | 15.71    | 18.20    | 25.52  |
> ImOV3D  | 45.51    | 19.53    | 28.02    | 31.02  |
>
> **Q2**：L59-61 \
> In response to the question regarding the data used during the inference phase, we confirm that in our experiments on the ScanNet and SUNRGBD datasets, our method strictly used only point cloud data for inference and did not utilize image information. Within our framework, although it may appear that images and point clouds are used concurrently during the inference phase, we would like to clarify that the images presented are actually pseudo images generated by our rendering module based on the input point cloud data. While these pseudo images are visually similar to real images, they do not contain real image data inputs. We have adopted the  previous baseline method (OV3DET and CoDA), it guarantees that our inference is exclusively based on point cloud data, aligning with the methodological rigor of using a single modality for input. You can see more detailed intermediate results of the pipeline visualized in Fig 2 and 3 of the global pdf response.\
>
> 【Dear Reviewer】
> After reviewing our rebuttal response, do you have any further questions or areas that require additional clarification? We place great importance on your opinions and hope to continue the discussion and address any remaining concerns you may have in the upcoming communication session.

---

> ### Author Response · Authors · 2024-08-12
>
> Dear Reviewer m5vv, thank you for your valuable comments. As the deadline is approaching, we kindly inquire whether our discussions have addressed your concerns. If you have any further questions, we would be happy to continue our conversation. If our response has resolved your concerns, we would greatly appreciate it if you could consider updating your score and providing feedback. Thank you again.

---

> > ### Comment · Reviewer_m5vv · 2024-08-13
> > **Post rebuttal**
> >
> > I appreciate the authors' clarifications.  Authors have addressed my concerns.

---

> > > ### Author Response · Authors · 2024-08-13
> > >
> > > Dear Reviewer m5vv，we are very pleased to have resolved your concern. Thank you for your affirmation, we hope to receive your support！

---

### Official Review · Reviewer_HQsS · 2024-07-13

**Soundness:** 3
**Presentation:** 2
**Contribution:** 3
**Rating:** 6
**Confidence:** 3

**Summary:**

The paper aims to develop an open-vocabulary 3D detector with the help of 2D data. The key idea is to lift 2D images to 3D point clouds by using metric monocular depth estimation models, combined with estimating the extrinsics and intrinsics. The 2D bounding boxes are lifted to 3D as well, and filtered based on some constraints like their usual object sizes that are provided by GPT-4. Additionally, the 3D pointclouds are rendered to 2D images and re-texturerized using control-net. Finally, the paper trains a detector on the lifted 2D image and pointcloud data to get a base detector in "pre-training" stage. Then, the pseudo labels are generated on real 3D point clouds and projected to 2D RGB images, and trained further for a few epochs. The paper compared against existing baselines like OV-3DET and outperform them.

**Strengths:**

- open vocabulary 3d detection is an important problem and the paper makes progress on that front
- the proposed pipeline of lifting 2d images to 3d images and 3d point clouds to 2d is interesting.
- i appreciate the through implementation details throughout the paper and the supplementary materials.

**Weaknesses:**

- the paper claims (even in the title) to make a 3D open vocabulary object detector ONLY from 2D images. However, the results indicate that such a detector is significantly worse than detectors that uses real point cloud data. I think the "only 2D images" can be misleading and the authors should consider removing/modifying it.
- While the writing for the most part is very clear, I am very unclear on the model architecture. The description seems to be missing, and from the figure it seems like the 2D images are processed by an open-vocabulary detector and 3D point clouds by a 3D detector and somehow their predictions are merged to generate the final output. The description of losses could also really benefit a lot from a more detailed description.
- As already acknowledged in the limitations, the proposed pipelines adds significantly more complexities over OV-3DET while improving the performance of their point cloud version by about 2% (although the increase of the 2D only version is significant).

**Questions:**

- Some more explanation on the model architecture and losses would help a lot

**Limitations:**

yes

---

> ### Author Rebuttal · Authors · 2024-08-07
>
> **Q1**：\
> We gladly accept the suggestion and are considering modifying it to “*How far can 2D images drive 3D open-vocabulary object detection?* ” We appreciate your advice, but our motivation is to explore how can we fully exploit information from 2D images and how far they can drive the performance of 3D understanding both in an adaptation-free setting and an adaptation setting. We are also open to any suggestions.\
> **Q2**：\
> For a better understanding, please refer to Figure 1 in the global pdf response. \
> 【2D and 3D Fusion Details】\
> (1) Seed point generation and feature extraction\
> In the 3D branch, K seed points are initialized from the 3D point cloud, each defined by its coordinates in 3D space (Kx3). These seed points are further enriched by feature extraction, where each point, in addition to its coordinates, also acquires additional feature representations $F_{pc} \in \mathbb{R}^{K \times F}$. \
> (2) Lifting 2D information to 3D\
> Using a 2D OV detector to identify objects on RGB images generates 2D bounding boxes and semantic labels. These 2D information are transformed into 3D rays through the camera matrix, with these rays originating from the seed points and pointing towards the center of the objects in 3D space. This process elevates the geometric cues from the 2D image into 3D space, aiding in the precise localization of objects. These lifted image features can be represented as $F_{img} \in \mathbb{R}^{K \times (3+F')}$.\
> (3) 2D and 3D Feature Fusion\
> During the feature fusion phase, the point cloud features $F_{pc}$ are concatenated with the image features $F_{img}$ to form the joint feature representation $F_{joint} \in \mathbb{R}^{K \times (3+F+F')}$. As a result, each seed point contains not only the geometric information in 3D space but also the semantic information integrated from the 2D image.\
> (4) Multi-Tower Architecture\
> We employ a multi-tower architecture to process the point cloud features and 2D image features separately. Each tower focuses on handling one type of input feature and balances the contributions of different modalities through a gradient fusion strategy. The img tower has a weight of 0.3, the point tower has a weight of 0.3, and the joint tower has a weight of 0.4, consistent with the backbone ImVoteNet.\
> 【Loss Details】\
> Finally, the loss function can be expressed as:
> $$
> \small
> L_{\text{total}} = L_{\text{loc}} +  \sum_{i} W_i \times \text{CrossEntropy}(\text{Cls-header}(F_{i}) \cdot F_{\text{text}})
> $$
>
> where $i$ represents different features, such as $\text{pc}$, $\text{img}$, $\text{joint}$. $W_i$ is the weight corresponding to feature $i$. $L_{\text{loc}}$ represents the original localization loss function used in ImVoteNet. $F_{\text{text}}$ denotes the feature extracted by the text encoder in CLIP.\
> **Q3**：\
> Our research paper demonstrates the immense potential of using a 2D-only setting. Through our approach, we primarily aim to address the question of how much 2D images themselves can enhance the performance of OV 3D object detection. In our work, this is not just a paper that solves existing problems but also a scientific exploration. We believe that the 2D-only version setting, as an important experimental conclusion, shows people that 2D images can also provide abundant information for 3D understanding. The current increase in the adaptation setting exceeds the improvements made by previous works, such as CoDA. Although our method increases the complexity of training, our ablation study has proven that every step in the training stage is crucial. However, in the inference stage, our scheme is no more complex than the existing OV3DET. In actual use, the complexity of our scheme does not affect the user experience. The latency remains comparable to OV3DET, and the additional memory cost is even smaller. To promote the reproducibility of research, we will also fully open-source our code for everyone to reproduce and further study.
>
> 【Dear Reviewer】After reviewing our rebuttal response, do you have any further questions or areas that require additional clarification? We place great importance on your opinions and hope to continue the discussion and address any remaining concerns you may have in the upcoming communication session.

---

> > ### Comment · Reviewer_HQsS · 2024-08-10
> >
> > Thank you for the response, especially the detailed description of the architecture (and the architecture diagram). I am still supportive of accepting this paper.

---

> > > ### Author Response · Authors · 2024-08-12
> > >
> > > Dear Reviewer HQsS，thank you for your continued support and positive feedback. We're glad the detailed description and architecture diagram were helpful. Thank you again！

---

### Official Review · Reviewer_TsJm · 2024-07-15

**Soundness:** 3
**Presentation:** 3
**Contribution:** 2
**Rating:** 5
**Confidence:** 4

**Summary:**

This paper presents a novel LiDAR-based open-vocabulary 3D detection model that relies solely on 2D images for training, without using any 3D annotations. During the training phase, a pre-trained monocular depth estimation model generates depth maps from 2D images, which are then projected into pseudo point clouds. ControlNet is used to render 2D pseudo images from these pseudo point clouds. The 3D detection model takes both pseudo 3D point clouds and pseudo images as input to predict 3D bounding boxes. During inference, the model uses ground truth point clouds and rendered 2D images to predict the 3D boxes. Additionally, the authors employ GPT to filter the pseudo annotations. The results demonstrate that this model achieves state-of-the-art performance on the SUNRGBD and ScanNet datasets. The authors have also released their code in the supplementary materials.

**Strengths:**

1. The proposed approach is novel as it only requires 2D images to generate pseudo point clouds and pseudo images for training the 3D detection model, which includes auto-generating pseudo 3D ground truth.
2. The model outperforms existing SOTA models on two datasets.
3. The paper is well-written with a detailed methodology, making it easy to understand the proposed method.
4. The authors provide the code, ensuring reproducibility.

**Weaknesses:**

The model achieves great performance in open-vocabulary 3D detection, but there are several concerns that need to be addressed:

1. Given that the inference input is only the point cloud, why is the training input limited to images? The training data also includes point clouds; why not utilize these as well?
2. 3D Pseudo Label Generation: The pseudo point cloud is noisy, making it challenging to generate accurate 3D pseudo ground truth from it. The paper lacks details on how the 3D boxes are fitted to the point cloud and the accuracy of the pseudo ground truth. Providing these details is crucial.
3. Complex Pipeline: The overall pipeline is complex. Could the image generation step be removed and the original image used instead? During inference, could both image and LiDAR data be used, as in OV-3DET?
4. Training Data Overlap: The paper assumes that the training data does not include LiDAR. Is there any overlap between the depth model training data and the detection dataset?
5. Minor issue: Does not cite ControlNet in the main paper.

**Questions:**

Please the authors address my questions presented in the weaknesses.

**Limitations:**

Not obvious limitation associated with broader societal impacts.

---

> ### Author Rebuttal · Authors · 2024-08-07
>
> **Q1**：\
> We focus on 2D images because real-world 3D point cloud data is not only limited in quantity but also has relatively few annotations. At the same time, we have observed that 2D image data is not only abundant in quantity but also rich in annotation information. Based on these observations, we wish to explore the potential of 2D images to enhance 3D understanding performance under the condition of having only 2D images.
>
> In the adaptation stage, we do indeed use real 3D point cloud data, but the main purpose is to reduce the differences between the model's performance across different data domains, that is, to minimize the domain gap. The reason we choose not to use point clouds and images simultaneously during the training phase is that the existing RGB-D paired datasets are small in scale. We do not want the model to be optimized only for these limited datasets during the adaptation stage, thereby restricting the model's generalization ability.
>
> Therefore, our approach aims to improve the performance of the 3D detection model by making full use of the rich 2D image data and utilizing the limited 3D point cloud data in the adaptation stage to optimize the model. This ensures that the model maintains efficient and accurate performance when facing real-world data, while also possessing good generalization capabilities.
>
> **Q2**：\
> 【***3D Pseudo Annotation Generation***】\
> Firstly, we determine the boundaries of the frustum based on the 2D bounding boxes, and then extract all points within the frustum from the 3D point cloud. The extracted point cloud may contain background points and outliers. To remove these points, we employ a clustering algorithm to analyze the point cloud, gathering points that belong to the same object together. Through the clustering results, we can identify and remove background points and outliers that do not belong to the target object, thus obtaining cleaner point cloud data. After removing non-target points, we calculate the 3D pseudo bounding box based on the remaining points. This includes determining the center position, size, and orientation of the pseudo box. Typically, the center position can be determined by the centroid of the point cloud, the size can be estimated by the boundary of the point cloud, and the orientation can be determined by the object's main axis or normal. Therefore, we can generate corresponding pseudo labels, including the position, size, and orientation information of the box. \
> 【***The Accuracy of the Pseudo Labels***】\
> Since the depth images Dmetric obtained from monocular depth estimation may contain noise, this can affect the accuracy of the 3D bounding boxes. To address this issue, we use a 3D bounding box filtering module to filter out inaccurate 3D bounding boxes. We have constructed a median object size database using GPT-4, by asking for the average length, width, and height of specific categories, defining a filtering criterion for comparing the size L, W, H of each object in the scene with the median size. If the ratio of the object's dimensions to the median size is within a range of the threshold T, the object is retained. We filter out bounding boxes that do not meet the size criteria before training. During inference, we remove categories that are semantically similar but have different sizes to prevent errors such as misidentifying a book as a bookshelf. From Table 3 in the paper, the quantitative results of this part can be seen. We will elaborate this more clearly in the main paper. \
> **Q3**：\
> 【***Answer questions about the complex pipeline***】\
> Understanding 3D scenes using only images is an incredibly challenging problem. That's why we've designed this component and the entire pipeline. Although it may appear complex at first glance, every part of it is essential. We haven't included any components to make it redundant or to add unnecessary complexity. Instead, this seemingly complex pipeline is carefully crafted to include everything necessary to effectively address the task. \
> 【***Regarding the possibility of removing image generation step***】\
> Please review the global reply.\
> 【***Inference process***】\
> The OV3DET operates on a point cloud only basis during the inference process, which is not what question mentioned as being able to input both image and point cloud simultaneously. On the contrary, our pipeline can incorporate images during the inference process. However, Tables 3 and 4 in the PDF indicate that using point clouds and images during inference leads to a decline in results. The reason is the domain gap between the rendered images from training and the real images from inference. \
> **Q4：**\
> In our paper, the training data for the depth estimation model comes from a mix of 12 datasets (e.g. NYU DepthV2,KITTI etc), excluding SUNRGBD and Scannet, which are used for detection. This ensures there is no overlap in the training data.\
> **Q5**：\
> Our apologies, we will cite "Controlnet" in the main paper subsequently.
>
> 【Dear Reviewer】After reviewing our rebuttal response, do you have any further questions or areas that require additional clarification? We place great importance on your opinions and hope to continue the discussion and address any remaining concerns you may have in the upcoming communication session.

---

> > ### Comment · Reviewer_TsJm · 2024-08-13
> >
> > Thank the authors for their response. It has successfully addressed my concerns. I would like to upgrade my voting of the paper. Please incorporate the suggested revisions in the next version of the manuscript.

---

> > > ### Author Response · Authors · 2024-08-13
> > >
> > > Dear Reviewer TsJm, we are grateful for your valuable feedback and are committed to incorporating the suggested revisions in the next version of our manuscript！Thank you for your support！

---

> ### Author Response · Authors · 2024-08-12
>
> Dear Reviewer TsJm, thank you for your valuable comments. As the deadline is approaching, we kindly inquire whether our discussions have addressed your concerns. If you have any further questions, we would be happy to continue our conversation. If our response has resolved your concerns, we would greatly appreciate it if you could consider updating your score and providing feedback. Thank you again.

---

### Official Review · Reviewer_teog · 2024-07-15

**Soundness:** 2
**Presentation:** 3
**Contribution:** 2
**Rating:** 5
**Confidence:** 3

**Summary:**

The paper introduces ImOV3D, a novel framework for open-vocabulary 3D object detection (OV-3Det) that learns exclusively from 2D images. The method addresses the scarcity of annotated 3D data by leveraging the wealth of annotations in 2D images. ImOV3D employs a pseudo-multimodal representation that bridges the gap between 2D images and 3D point clouds, enabling effective knowledge transfer. The framework converts 2D images to pseudo-3D point clouds using monocular depth estimation and vice versa, integrating 2D semantic information with 3D spatial characteristics. Extensive experiments on SUNRGBD and ScanNet datasets demonstrate significant performance improvements over existing methods.

**Strengths:**

1.The approach of using only 2D images to train a 3D object detector is highly innovative and addresses a critical bottleneck in the field.

2.The framework’s ability to convert 2D images into 3D point clouds and back into 2D images provides a robust method for integrating multimodal data.

3.The paper includes extensive experiments and ablation studies on benchmark datasets, showcasing significant improvements over state-of-the-art methods.

4.The framework achieves impressive results without ground truth 3D data, and even better performance with minimal 3D data for fine-tuning.

**Weaknesses:**

Real-World Performance: How does ImOV3D perform when applied to real-world 3D data beyond the benchmark datasets used in the paper?

Adaptability: Can the model be easily adapted or fine-tuned for different types of 3D environments, such as outdoor scenes or highly dynamic environments?

Resource Requirements: What are the computational resources required for training and inference? Can the method be efficiently scaled for use in large-scale applications? What is the latency?

**Questions:**

Please see the weakness.

**Limitations:**

Please see the weakness.

---

> ### Author Rebuttal · Authors · 2024-08-07
>
> **Q1**：\
> While our method, ImOV3D, has been primarily designed and evaluated for indoor scenes, we recognize the importance of assessing its performance on real-world 3D data beyond the benchmark datasets mentioned in our paper. To address this, we conducted experiments on the KITTI dataset, which consists of outdoor scenes, to evaluate how well our method generalizes to different datasets.
> ##### Table 1: Results from the Pretraining Stage Comparison of Open Vocabulary 3D Object Detection Performance on KITTI Moderate Difficulty Level
>
> | **Method**   | **Car** | **Pedestrian** | **Cyclist** | **mAP\@0.25** |
> |--------------|---------|----------------|-------------|--------------|
> OV-VoteNet | 11.32 | 8.64 | 8.08 | 9.35 |
> OV-3DETR | 13.68 | 12.08 | 8.56 | 11.44 |
> OV-3DET | 12.92 | 6.24 | 8.52 | 9.22 |
> ImOV3D | 20.42 | 11.24 | 14.96 | 15.54 |
>
> The results indicate that our method, while showing strong performance on indoor datasets, also exhibits a reasonable level of robustness and adaptability when confronted with outdoor scenes. Compared to other baselines, our mAP\@0.25 is 4.1% higher. The pretraining images encompass both indoor and outdoor scenes, hence allowing us to naturally generalize to outdoor scenes.
>
> **Q2**：\
> We fine-tuned our model using real point cloud data, and the results show that our method outperforms OV3DET by 5.5% in the mAP\@0.25, indicating a significant enhancement in performance after fine-tuning (compared to the data in Table 1 of Q1). However, the relatively limited categories in outdoor scenes restrict the full potential of the open-vocabulary detector. Moreover, the differences between indoor and outdoor data also affect the outcomes. More refined adjustments and designs tailored for outdoor scenarios, our results could further improvement.
> ##### Table 2: Results from the Adaptation Stage Comparison of 3D Object Detection Performance on KITTI Moderate Difficulty Level
>
> | **Method**   | **Car** | **Pedestrian** | **Cyclist** | **mAP\@0.25** |
> |--------------|---------|----------------|-------------|--------------|
> OV-3DET | 42.65    | 15.71    | 18.20    | 25.52  |
> ImOV3D  | 45.51    | 19.53    | 28.02    | 31.02  |
>
>
> **Q3**：\
> 【***Computational Resources***】 In our research project, the model's pre-training stage utilized 8 NVIDIA GeForce RTX 3090 GPUs, taking 72 hours, while the adaptation stage required only 12 hours, with both stages demanding approximately 16GB of GPU memory per GPU with a batch size of 12. 0.2s per scene on average on the scannet and sunrgbd dataset. \
> 【***Scalability and Latency***】Our method is designed with scalability, suitable for large-scale applications. Its core strength lies in the flexibility of its backbone network, which can be seamlessly replaced with the latest efficient architectures, thereby further enhancing efficiency with future technological advancements.
>
> 【Dear Reviewer】After reviewing our rebuttal response, do you have any further questions or areas that require additional clarification? We place great importance on your opinions and hope to continue the discussion and address any remaining concerns you may have in the upcoming communication session.

---

> ### Author Response · Authors · 2024-08-12
>
> Dear Reviewer teog, thank you for your valuable comments. As the deadline is approaching, we kindly inquire whether our discussions have addressed your concerns. If you have any further questions, we would be happy to continue our conversation. If our response has resolved your concerns, we would greatly appreciate it if you could consider updating your score and providing feedback. Thank you again.

---

### Official Review · Reviewer_MLwT · 2024-07-28

**Soundness:** 3
**Presentation:** 3
**Contribution:** 3
**Rating:** 7
**Confidence:** 5

**Summary:**

This work addresses the problem of the lack of 3D data and attempts to train open-vocabulary point-based 3D detection models solely with 2D images to avoid using 3D data and annotations. It proposes a pseudo data generation pipeline for this purpose, which generally follows a paradigm of estimating depth, projecting 2D to 3D, and rendering back from 3D to 2D to obtain 2D-3D paired pseudo data. Experiments show that models trained with the generated data achieve better results than previous methods.

**Strengths:**

1. The motivation of this paper is reasonable.
2. The proposed method is needed in this field and indeed contributes to addressing the 3D data issue.
3. The experimental results, to some extent, show that the proposed method is effective.

**Weaknesses:**

My major concerns lie with the unclear motivation and implementation of some components.

1. Intuitively, using a fixed camera intrinsic causes inaccurate projection. This work does not propose specific designs for addressing this problem nor discuss the limitations of using fixed intrinsics.
2. The reasons for using Formula (1) are not explained.
3. The method for lifting 2D bounding boxes to 3D, as described from L161-L163, is unclear. Did the authors use the minimum and maximum depth of the four corners of the 2D bounding boxes to generate the frustum? How is the orientation of the 3D bounding boxes determined? How is the gap addressed between the pseudo 3D bounding boxes, which are oriented, and the 3D ground truth bounding boxes in ScanNet, which are axis-aligned?
4. For the pre-training stage, why not use the original 2D images directly for training, considering that there is still another adaptation training stage to handle the domain gap between images from real point clouds and pseudo point clouds? In other words, why not use the adaptation training stage to address the gap between real images and images from real point clouds, thereby skipping the process of generating pseudo images from pseudo point clouds?
5.  As the key contribution of this work is the data generation pipeline, there should be more qualitative results for each step to demonstrate how this pipeline truly works. In the rebuttal phase, I suggest the authors show the results (e.g., intermediate results in Figure 2, point clouds with pseudo 3D bounding boxes can be shown using Open3D) using the first image of scene0000_00 in ScanNet and the first image of scene0012_01 in ScanNet as examples.

The data generation pipeline itself is typical and lacks novelty (e.g., similar work like SpatialGPT exists). It is difficult to fully understand the implementations and evaluate the performance of the pipeline solely through the manuscript. Nevertheless, this work involves significant engineering effort and tricks and contributes to the 3D perception community. If the code is open-sourced, I suggest accepting the paper.

**Questions:**

See the weakness section.

**Limitations:**

See the weakness section.

---

> ### Author Rebuttal · Authors · 2024-08-07
>
> **Q1**：\
> In depth estimation research, using fixed camera intrinsics is common practice [1][2]. While we've followed this approach, we recognize it may lead to inaccurate bounding box sizes. To address this, we've implemented a scale filter using GPT-4, adjusting bounding boxes based on empirically determined object sizes from LLMs. This correction has significantly improved performance. As shown in Table 3, the first row presents results without bounding box correction, and the second row shows improved performance with the filtering strategy: mAP0.25 increased by 1.65% on the SUNRGBD dataset and by 1.27% on the ScanNet dataset. Thank you for your valuable feedback; we will include a discussion on this limitation in the paper.\
> **Q2**：\
> The motivation for using this formula is explained in the main text from L143 to 150. We explore Equation (1) in depth within paper, utilizing Rodrigues' rotation formula [3]. It calculates the rotation matrix $R$ for a rotation by $\theta$ (less than $180^\circ$) around the axis defined by the unit vector $\hat{n}=(n_x,n_y,n_z)$. The formula is defined as:
> $$R=I+(\sin\theta)N+(1-\cos\theta)N^2$$
> Where $I$ is the identity matrix, and the skew-symmetric matrix $N$ for the unit vector $\hat{n}$ is constructed as:
> $$N=\begin{bmatrix}0&-n_z&n_y;&n_z&0&-n_x;&-n_y&n_x&0\end{bmatrix}$$
> To align unit vector $N_{pred}$ with unit vector $Z_{axis}$, we start from the definitions of the inner product $N_{pred} \cdot Z_{axis} = \cos\theta$ and cross product $v = N_{pred} \times Z_{axis} \Rightarrow v = \sin\theta\hat{n}, |v| = \sin\theta$.
> Accordingly, define:
> $$K\stackrel{\mathrm{def}}{=}(\sin\theta)N= \begin{bmatrix}0&-v_z&v_y;&v_z&0&-v_x;&-v_y&v_x&0\end{bmatrix}$$
> Based on rotation formula, we have:
> $$R=I+K+\frac{1-\cos\theta}{\sin^2\theta}K^2 $$
> $$=I+K+K^2\frac{1-N_{pred}\cdot Z_{axis}}{||v||^2} $$
> **Q3**：\
> 【**Generating a Visual Frustum**】 \
> We utilize the four corner points of a 2D bounding box, along with the minimum and maximum depth values from the depth image, to create a visual frustum. Specifically, we first convert the 2D image coordinates into 3D spatial coordinates using the intrinsic matrix. Then, based on the depth information, we calculate the actual positions of these corner points in 3D space, thereby generating the visual frustum. Within the visual frustum, we extract the corresponding point cloud data. To ensure the stability and efficiency of the calculations, we perform downsampling on the point cloud data. Subsequently, we apply the DBSCAN (Density-Based Spatial Clustering of Applications with Noise) clustering algorithm to the extracted point cloud data to remove outliers and noise, retaining only the main point cloud clusters.\
> 【**3D Bounding Box Orientation**】\
> By applying Principal Component Analysis (PCA) to the main point cloud clusters, we calculate the main orientation angle (yaw) of the point cloud. Specifically: PCA analysis is conducted on the X-Y plane of the point cloud data to obtain the principal component direction vector. The main orientation angle (yaw) is calculated based on the principal component direction vector. The point cloud is rotated using a rotation matrix to align the main direction with the X-axis. The range of the bounding box after rotation alignment is calculated. The bounding box is then rotated back to its original direction to obtain the final 3D bounding box position and orientation. We construct the final 3D bounding box description based on the calculated center point coordinates, dimensions (width, depth, and height), and orientation angle (yaw).\
> 【**Regarding the Orientation of Bounding Boxes in the ScanNet Test Dataset**】\
> The official bounding boxes provided by the ScanNet dataset are axis-aligned, in our research, we have adopted the baseline method (OV3DET and CoDA), following the common practices established by previous studies, using Principal Component Analysis (PCA) to determine the orientation of the bounding boxes. In all our experiments, we uniformly used Oriented Bounding Boxes for testing, having standardized the representation of bounding boxes, thus eliminating any discrepancies in bounding box orientation. At lines L162-163, we have provided a description pertinent to the issue raised. We will elaborate more on this in the paper to ensure clarity. \
> **Q4**：\
> As mentioned in the global reply.\
> **Q5**：\
> Thank you for your suggestion, we have followed your request and displayed the visualization of each step in Fig2 and Fig3 of the global response PDF.\
> **Q6**：\
> 【**About lack novelty**】\
> Our data generation pipeline is meticulously designed, and we do not agree that similar work implies a lack of novelty in our approach. The works you mentioned, such as Spatial-RGPT and Spatial VLM, are considered concurrent under the NeurIPS Policy, which treats works published within two months of submission as being from the same period. Our task design and complete pipeline differ significantly, particularly in our focus on 3D detection and detailed descriptions of 3D bounding boxes, including orientation and size. During the rebuttal phase, we will provide more detailed visualizations of the pipeline in the global PDF, Fig 2 and 3, to better illustrate the entire process. We are committed to open-sourcing our code to enable others to explore further applications based on our work, fostering development and innovation within the community.
>
> 【Dear Reviewer】We value your feedback and look forward to discussing any additional issues in the next communication phase.\
> **References** \
> [1]Pan Ji, Runze Li, Bir Bhanu, Yi Xu, "MonoIndoor: Towards Good Practice of Self-Supervised Monocular Depth Estimation for Indoor Environments"\
> [2]Vitor Guizilini, Igor Vasiljevic, Dian Chen, Rares Ambrus, Adrien Gaidon, "Towards Zero-Shot Scale-Aware Monocular Depth Estimation"\
> [3]Richard M. Murray, Zexiang Li, S. Shankar Sastry, pp. 26-28,"A Mathematical Introduction to Robotic Manipulation"

---

> > ### Comment · Reviewer_MLwT · 2024-08-10
> >
> > I thank the authors for addressing some of my concerns and suggest that these details be included in the final paper.
> >
> > As a reminder, according to the NeurIPS 2024 policy on “Contemporaneous Work”: “For the purpose of the reviewing process, papers that appeared online within two months of a submission will generally be considered ‘contemporaneous’ in the sense that the submission will not be rejected on the basis of the comparison to contemporaneous work.” SpatialVLM (CVPR 2024), which appeared in January 2024, is not considered concurrent work. Therefore, I would like the authors to further clarify how SpatialVLM differs “significantly.”

---

> > > ### Author Response · Authors · 2024-08-12
> > >
> > > Dear Reviewer,
> > >
> > > Thank you for your correction. We realized that we misunderstood the NeurIPS-related policy in our previous expression when comparing with SpatialVLM, which led to a lack of clear communication about the main differences between the two. In the revision, we will discuss the specific differences with SpatialVLM in more detail and highlight the key points.
> > >
> > > Below are the important differences between ImOV3D (ours) and SpatialVLM:
> > >
> > > 1. **Innovation Focus**
> > >    - Compared to SpatialVLM, although both involve the generation of pseudo point clouds and labels, the innovation of ImOV3D goes beyond this. The goal of ImOV3D is to create pseudo multimodal data for open-vocabulary 3D object detection. This process also includes a rendering stage, where we leverage a large number of 2D images to learn a general and color-free renderer that converts geometric point clouds into images. This allows us to generate high-quality pseudo images, which not only enhances the richness and reliability of the data but also improves our performance in 3D object detection tasks.
> > >
> > > 2. **Horizontal Plane Mask Extraction**
> > >     - For the horizontal plane mask, ImOV3D extracts it through the normal map of the RGB image. In this normal map, the red, green, and blue channels correspond to the three different components of the normal vector. A darker green channel indicates a significant vertical component, allowing us to effectively extract the horizontal plane mask by setting a threshold for the green channel. Compared to the segmentation model used by Spatial VLM, which relies on identifying specific semantic labels such as "floor" or "table" to extract the horizontal plane mask, this method also has issues with stability and reliability because it depends on manually defined semantic categories that are not guaranteed to appear in all images. Our method directly extracts accurate geometric information from the image, providing the normal direction for each pixel point without relying on specific semantic labels. This direct extraction of normal information is widely applicable and not limited to specific scenes or objects.
> > >
> > >     - Additionally, Spatial VLM does not provide implementation details regarding the extrinsics estimation nor open source the code, making it difficult for us to perform a detailed methodological comparison. Our method mathematically ensures that we identify the minimal angular rotation among all possible transformations capable of aligning the normals, thereby avoiding unnecessary rotations and preserving the geometric integrity of the depth point cloud, ensuring its broad applicability and reliability.
> > >
> > > 3. **3D Bounding Box Quality Control**
> > >    - SpatialVLM does not have quality control for 3D Bbox when generating 3D data. Due to the accumulation of errors from point cloud lifting, camera intrinsic and extrinsic parameters, 2D bounding boxes, and 3D clustering, quality control is crucial for high-quality box labeling. ImOV3D, focusing on 3D detection, introduces more quality control strategies, such as a 3D bounding box filtering module. By using the median size of objects generated by GPT-4, pseudo 3D bounding boxes are filtered to ensure the accuracy and reliability of the data. This strategy can effectively reduce the impact of noise data and improve the quality of the final detection results.
> > >
> > > 4. **Bounding Box Generation Method**
> > >    - During the process of generating 3D bounding boxes, we have chosen a more universal approach, employing bounding boxes instead of segmentation. Particularly near the edges, segmentation can still be affected by outlier points, which does not resolve the issue but rather imposes a higher demand for 2D annotations. We utilize the 2D bounding boxes provided in the 2D dataset to construct the frustum, ensuring that the generation of 3D bounding boxes is both accurate and efficient.
> > >
> > > Thank you for your professional and serious reply. If you have any other questions, we are happy to communicate with you!

---

> > > > ### Comment · Reviewer_MLwT · 2024-08-12
> > > >
> > > > Thanks for the reply!
> > > >
> > > > I have some additional questions regarding the implementations:
> > > >
> > > > 1. When generating the visual frustum, if only the four corners are used, is it possible that the 3D locations of the corners of the 2D bounding box, having larger depth values than the object, could result in inaccurate cropping of the object point clouds? This concern arises because the 2D corners typically do not rest directly on the object.
> > > > 2. When using DBSCAN, over-segmentation and under-segmentation often occur because the object point clouds are incomplete, especially when projected from a single view. Did the authors encounter these issues, and how did they address them?
> > > > 3. For horizontal plane segmentation, the authors use a normal map, but this method appears to result in false segmentation, as evidenced by Figures 2 and 3 in the rebuttal PDF. Could the authors provide more details on this issue?

---

> > > > > ### Author Response · Authors · 2024-08-12
> > > > >
> > > > > Thank you for your reply! Below are our detailed answers to your questions:
> > > > >
> > > > > **Q1:**
> > > > > When relying solely on the four corners of a 2D bounding box to estimate the 3D position of an object, we indeed encounter the issue you mentioned: it results in bounding boxes that are overly large and fail to accurately enclose the object's point cloud. To address this, we first extract 3D points within a frustum-shaped 3D bounding box. We then apply clustering techniques to remove background noise and outliers, thereby improving the precision of the bounding box relative to the point cloud. It's important to note that outlier points can inevitably occur, whether using bounding boxes or segmentation. However, while clustering might seem concerning due to the potential impact of these outliers, in practice, it does not pose a significant issue. This is because we follow up with a filtering process designed to further ensure the quality of the labels. Utilizing the clustered points, we accurately determine the object's center, dimensions, and orientation, which are crucial for refining the training of the 3D detector.
> > > > >
> > > > > **Q2:**
> > > > > To address the challenges of over-segmentation and under-segmentation inherent in the DBSCAN algorithm, we began by systematically experimenting with a range of hyperparameters to identify the most effective combination for our dataset. Throughout this process, we closely monitored the sizes of the bounding boxes (bboxes) generated after each DBSCAN iteration, comparing them meticulously with object dimensions informed by GPT-4's common sense estimates. Our goal was to select hyperparameters that would maximize the retention of bounding box counts while ensuring that the dimensions closely aligned with the actual sizes of the objects.\
> > > > >  After extensive trials and iterative refinements, we determined that setting the hyperparameters to eps=0.3 and min_samples=100 yielded the best results. These settings consistently produced optimal outcomes in our experiments, satisfying our criteria for 3D bounding boxes and ensuring that their dimensions closely matched the true object sizes. Following this, we implemented a 3D bounding box size filtering process based on these selected parameters, further enhancing the quality of both the point clouds and the bounding boxes generated post-clustering by the DBSCAN algorithm. We haven't identified significant impacts at this time, and we will release all the relevant code to benefit the community.
> > > > >
> > > > > **Q3:**
> > > > > For horizontal plane segmentation, we set a relatively high threshold (threshold=230) to extract the horizontal plane regions from the normal map, corresponding to the darker green areas. However, the normal estimator may occasionally misclassify a few pixels as part of the horizontal plane. To mitigate the impact of these “false” planes, we implemented the following approach (as described in the paper from L146 to 149):
> > > > > After obtaining the horizontal mask, we extracted the normal vector for each pixel, \(N_i = (N_x, N_y, N_z)\). We then calculated the predominant normal vector for the horizontal plane, \(N_pred} = Cluster(N_i)\, using a clustering method to select the dominant cluster. This approach identifies the normal vector that best represents the overall horizontal plane by grouping similar vectors. Thus, even if individual pixel normals are slightly off, as long as the majority are accurate, the clustering result will generally yield a reliable normal vector for the horizontal plane.

---

> > > > > > ### Comment · Reviewer_MLwT · 2024-08-12
> > > > > >
> > > > > > Thanks.
> > > > > >
> > > > > > I suggest the authors add more detailed limitations with potential solutions of the method in the revised paper for future improvement from the community.

---

> > > > > > > ### Author Response · Authors · 2024-08-13
> > > > > > >
> > > > > > > Dear Reviewer,
> > > > > > >
> > > > > > > Thank you for your valuable suggestions. We agree to include a detailed discussion on the limitations of our method in the revised manuscript, and in addition, we will open source all the code to facilitate further research and community contributions!

---

> > > > > > > > ### Comment · Reviewer_MLwT · 2024-08-13
> > > > > > > >
> > > > > > > > Cool. Now I raise my score to 7.

---

> > > > > > > > > ### Author Response · Authors · 2024-08-13
> > > > > > > > >
> > > > > > > > > Thank you for your support! We are delighted that our paper has received your approval.

---

### Author Rebuttal · Authors · 2024-08-07

We are deeply grateful for the professional reviews and valuable feedback from all the reviewers. Reviewer MLwT acknowledged the motivation behind our research and its contribution to addressing the 3D data issue; Reviewer teog praised the innovation of our approach and the robust method for integrating multimodal data; Reviewer TsJm appreciated the novelty of our method that generates pseudo point clouds and images using only 2D images; Reviewer HQsS affirmed the detailed implementation and supplementary materials we provided; Reviewer m5vv noted the significant improvements achieved by our method on the SUNRGBD and ScanNet datasets. We value every piece of feedback and have responded to each reviewer's questions individually. More additional experiments were also conducted during the rebuttal phase to support our proposed method, as suggested by the reviewers (given in the PDF file).

In response to the question raised by **Reviewer MLwT Q4** and **Reviewer TsJm Q3** regarding "***whether the image generation step can be removed and original images can be used instead***," we provide the following reply:

（1）Our OV 3D detector utilizes only point cloud input during both the pre-training stage and the adaptation stage. These point clouds contain only geometric information and do not include color information. We need and can only use pseudo point clouds to train a general point cloud renderer that is applicable to various scenarios. This way, we can generate images with only point cloud inputs, thus leveraging the capability of multimodal detection.\
（2）For such a point cloud renderer, it is extremely challenging to narrow the gap between the images rendered from real point clouds and real images. In contrast, the difference between the images rendered from real point clouds and pseudo point clouds is not significant. Therefore, we choose to use pseudo images in the pre-training stage. This way, during the adaptation training and inference stages, the image branch can migrate well since the quality gap between images rendered from real point clouds and those used in training is small.\
（3）If we use real images during the pre-training stage, in a zero-shot setting, the large gap between real images and images rendered from real point clouds will lead to a failure in migration. Even if a small amount of RGB-D data is used for adaptation, it is still difficult to bridge the gap between real images and rendered images, causing the 2D branch to fail. \
（4）In the global response of the PDF: The first row of Table 1 and Table 2 shows the results of consistently using the renderer during pretraining, adaptation, and inference stages. This indicates that maintaining consistent data representation throughout the process can yield optimal performance.

The second row of Table 1 and Table 2 in the PDF's global response reflects the decline in mAP\@0.25 on both datasets when real images are used during training and the renderer is used during inference. This reveals a significant difference between real images and the pseudo images generated by the renderer, a discrepancy that affects the model's ability to transfer knowledge.

Similarly, the second row of Table 3 and Table 4 also shows a trend of performance decline when the renderer-generated pseudo images are used during training and real images are used during inference. This further confirms the negative impact of using different data representations during training and inference stages.

These results suggest that in order to maintain the stability and consistency of the model across different stages, we cannot use different image sources during training and inference. The pseudo images generated by the renderer are indispensable throughout the process because they provide a representation close enough to real images, ensuring the model can learn and transfer knowledge effectively. Although a limited amount of real data is introduced during the adaptation phase to reduce the gap between real and pseudo images, this approach still fails to surpass the performance achieved by using the renderer in all stages. This underscores the importance of the rendering step throughout the process and that it cannot be omitted.

【Dear Reviewer】
After reviewing our rebuttal response, do you have any further questions or areas that require additional clarification? We place great importance on your opinions and hope to continue the discussion and address any remaining concerns you may have in the upcoming communication session.

---

### Decision · Program_Chairs · 2024-09-25

**Decision:**

Accept (poster)

**Comment:**

The paper aims to develop an open-vocabulary 3D detector with the help of 2D data. The key idea is to lift 2D images to 3D point clouds by using metric monocular depth estimation models, combined with estimating the extrinsics and intrinsics. Finally, the paper trains a detector on the lifted 2D image and point-cloud data to get a base detector in "pre-training" stage. Then, the pseudo labels are generated on real 3D point clouds and projected to 2D RGB images, and trained further for a few epochs. The paper compared against existing baselines like OV-3DET and outperform them. The initial key concerns are the clarity in many components, more real-world 3D dataset experiments are needed, overall pipeline complexity. In the rebuttal, the authors give more details of the key components and show additional experiments on KITTI dataset. The final ratings are two borderline accept, one weak accept, and one accept. Hence, the AC recommend accept.